# Superconductivity in a van der Waals layered quasicrystal

Yuki Tokumoto [1] ✉, Kotaro Hamano[1], Sunao Nakagawa[1], Yasushi Kamimura [1], Shintaro Suzuki [2], Ryuji Tamura [3] & Keiichi Edagawa [1] ✉

Van der Waals layered transition-metal chalcogenides are drawing significant attention owing to their intriguing physical properties. This group of materials consists of abundant members with various elements, having a variety of different structures. However, they are all crystalline materials, and the physical properties of van der Waals layered quasicrystals have never been studied to date. Here, we report on the discovery of superconductivity in a van der Waals layered quasicrystal of $Ta_{1.6}Te$. The electrical resistivity, magnetic susceptibility, and specific heat of the quasicrystal unambiguously validate the occurrence of bulk superconductivity at a transition temperature of ~1 K. This discovery can promote new research on assessing the physical properties of novel van der Waals layered quasicrystals as well as two-dimensional quasicrystals; moreover, it paves the way toward new frontiers of superconductivity in thermodynamically stable quasicrystals.

In 1984, Shechtman et al. discovered an exotic solid phase in a rapidly quenched Al–Mn alloy whose electron diffraction pattern was sharp and exhibited ten-fold rotational symmetry[1]. Soon after this discovery, a new classification scheme for solids was proposed, with the aforementioned phase denoted a quasicrystal (QC)[2–4], which was defined by quasiperiodic translational order and crystallographically forbidden rotational symmetries such as five-, ten-, and twelve-fold symmetries. In 1992, the International Union of Crystallography modified the definition of crystals in order to incorporate QCs into crystals[5], and revised the classification scheme and definition of QCs accordingly. Now, all solids with long-range ordered atomic arrangements are called crystals. This includes 'aperiodic crystals', whose long-range order is not periodic but quasiperiodic. QCs are defined as aperiodic crystals that are neither incommensurate modulated structures nor aperiodic composite crystals. One significant difference between QCs and periodic crystals is that QCs are allowed to have a crystallographically forbidden rotational symmetry, which is, though, not a necessary but a sufficient condition for a QC in today's context. Extensive studies over the last 40 years have shown that QCs are not exceptional but ubiquitous, as they have been formed in several binary and ternary

systems. Of the 70 QCs reported to date, approximately 40 are thermodynamically stable[6].

In 1998, Conrad et al. discovered a unique stable QC phase of $Ta_{1.6}Te$, which exhibited dodecagonal symmetry[7]. This new QC has several notable features that have not been observed to date. First, this phase is the only reported transition-metal chalcogenide QC. Second, this is the only QC of a van der Waals (vdW) layered material; its structure is realized through periodic stacking of two-dimensional dodecagonal QCs. The structures of this dodecagonal QC phase and the related crystal approximant (CA) phases have been examined by X-ray diffractometry (XRD), electron diffractometry, and high-resolution transmission electron microscopy[7–11]. Moreover, a few layers of the dodecagonal QC, which can be regarded virtually as a two-dimensional QC, have recently been isolated using a standard exfoliation technique[8]. However, the number of studies on this phase is still limited because conventional solidification techniques such as arc-melting and induction melting, which are usually used for the fabrication of QC alloys, cannot be applied since the melting temperature of Ta is much higher than the boiling temperature

[1]Institute of Industrial Science, The University of Tokyo, Tokyo 153-8505, Japan. [2]Department of Physical Science, Aoyama Gakuin University, Kanagawa 252-5258, Japan. [3]Department of Materials Science and Technology, Tokyo University of Science, Tokyo 125-8585, Japan. ✉e-mail: tokumoto@iis.u-tokyo.ac.jp; edagawa@iis.u-tokyo.ac.jp

of Te. Moreover, experimental studies on elucidating the physical properties of the $Ta_{1.6}Te$ dodecagonal QC phase have not been reported.

Recently, vdW layered crystals of transition-metal chalcogenides have attracted much attention and have been studied extensively owing to their fascinating physical properties[12–16]. In the early stage of QC research, a few studies reported on the observations of super-conductivity in icosahedral QCs (i-QCs) of Al−Cu−(Mg,Li)[17,18]. However, convincing evidence for bulk superconductivity in QCs, indicated by zero resistivity, Meissner effect, and heat capacity jump, has only recently been presented; in 2018, Kamiya et al. observed the bulk superconductivity in a metastable i-QC of Al−Zn−Mg[19]. This i-QC phase, which was fabricated by a melt-spinning technique, exhibited a weak-coupling superconductivity at a transition temperature $T_c$ of ~0.05 K. In the present study, we fabricate the $Ta_{1.6}Te$ vdW layered QC phase by reaction sintering and measure the electrical resistivity, magnetic susceptibility, and specific heat down to low temperatures. The results

categorically confirm the occurrence of bulk superconductivity in a thermodynamically stable QC for the first time at $T_c \approx 1K$.

## Results

### Sample fabrication and characterization

Polycrystalline samples of the $Ta_{1.6}Te$ dodecagonal QC were fabricated by reaction sintering, which were powdered and then subjected to electron diffractometry and XRD analyses (see Methods section). The electron diffraction pattern of the synthesized sample (Fig. 1a) shows an arrangement of sharp spots with dodecagonal symmetry, verifying the formation of a dodecagonal QC phase. This pattern is essentially identical to that previously reported for the $Ta_{1.6}Te$ dodecagonal QC phase[7]. A powder XRD profile of the sample was acquired using Cu Kα radiation (Fig. 1b); the Cu Kα$_2$ component was removed numerically, yielding a profile that corresponded to the Cu Kα$_1$ component (wavelength: 1.5405 Å). A comparison of this profile with those of the crystalline phases of $TaTe_2$ and Ta−the materials used for the reaction

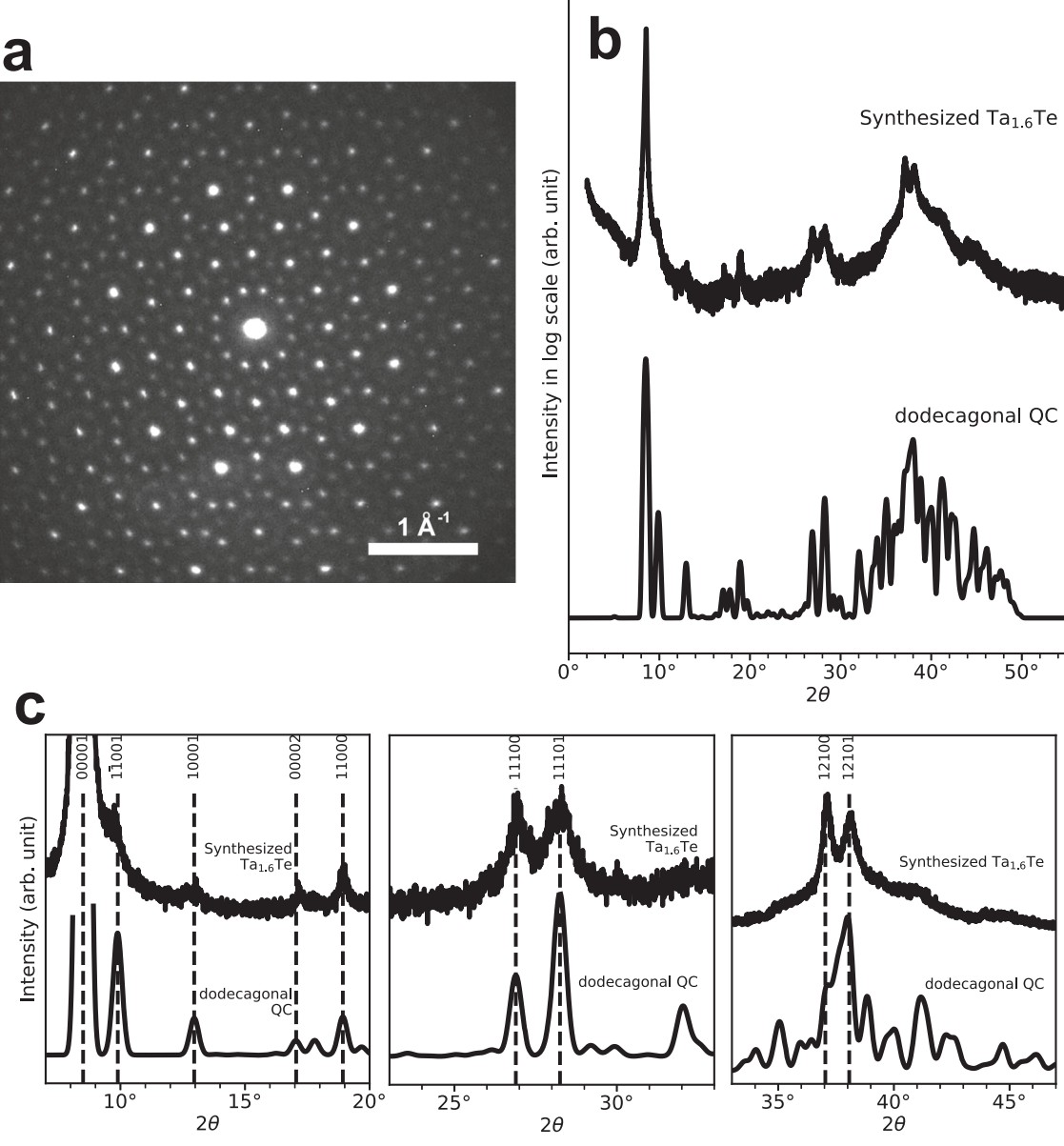

**Fig. 1 | Electron and X-ray diffraction data. a** Electron diffraction pattern of the synthesized $Ta_{1.6}Te$. **b** Experimentally obtained powder XRD profile of the synthesized $Ta_{1.6}Te$ and the calculated profile for the dodecagonal QC phase.

**c**, Magnified version of the data shown in **b**. The principles and procedures underlying the calculations are detailed in Supplementary Information Section 1.2.

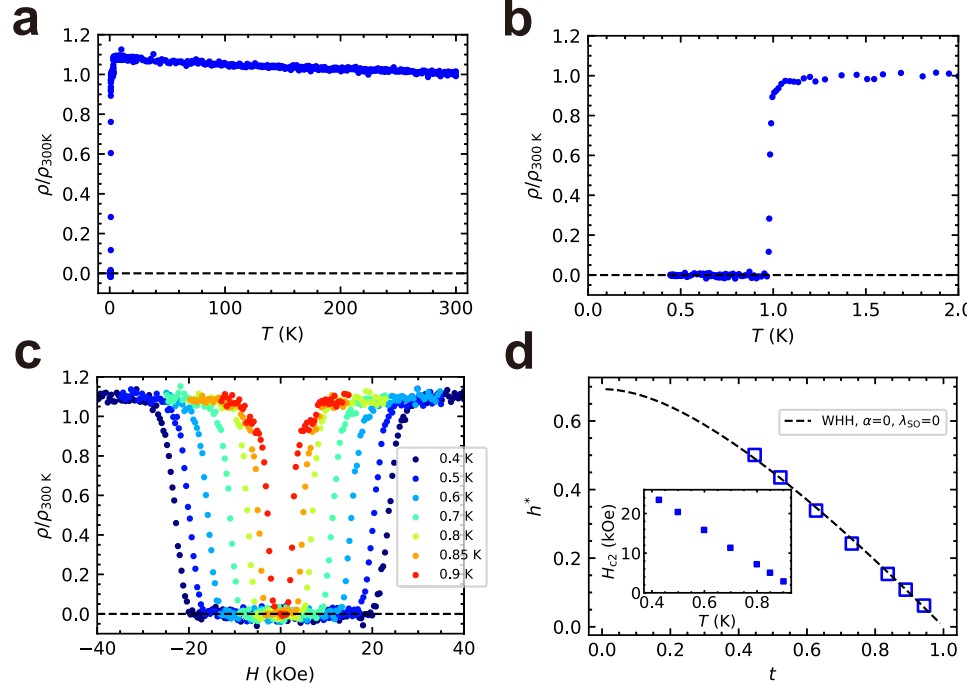

**Fig. 2 | Electrical resistivity measurements. a** Temperature dependence of the electrical resistivity normalized by the value at $T = 300$ K, i.e., $\rho/\rho_{300\,\mathrm{K}}$. **b** Temperature dependence of $\rho/\rho_{300\,\mathrm{K}}$ magnified at temperatures below 2 K. **c** Magnetic field dependence of $\rho/\rho_{300\,\mathrm{K}}$ at different temperatures. **d** Temperature dependence of the upper critical field $H_{c2}$ analyzed by plotting $h^* = -H_{c2}/\left[T_c\left(\frac{dH_{c2}}{dT}\right)_{T=T_c}\right]$ against $t = T/T_c$. Open squares represent the experimentally obtained data, and the dashed curve represents the Werthamer–Helfand–Hohenberg fit with $\alpha = 0$ and $\lambda_{SO} = 0$. The inset shows the temperature dependence of $H_{c2}$.

sintering–revealed no traces of these phases in the acquired profile, indicating the completion of the reaction and formation of another phase (Supplementary Fig. 1). However, because the powder XRD profile of the Ta$_{1.6}$Te dodecagonal QC phase has not been previously reported, it was constructed based on that of the CA phase of Ta$_{97}$Te$_{60}$, whose structure was previously determined through single-crystal XRD measurements[10]. In general, by introducing phason strain to a QC structure, a CA structure can be obtained. Using this relationship, the powder XRD profile of a QC phase can be calculated from that of its CA phase. The principles and procedures underlying the calculation are detailed in Supplementary Information Section 1.2 (Supplementary Figs. 3–9) and Supplementary Data 1 and 2. The acquired and constructed profiles were compared (Fig. 1b and the magnified profiles in Fig. 1c). Although the peaks in the acquired profile were not adequately resolved in the $2\theta$ region of 33–47°, the overall agreement was satisfactory. Notably, comparing the acquired XRD profile with those of the Ta$_{21}$Te$_{13}$ and Ta$_{97}$Te$_{60}$ CA phases showed clear discrepancies (Supplementary Fig. 9). The peak indices shown in Fig. 1c refer to the five basis vectors $\mathbf{a}_i^* = a^*(\cos(\frac{2\pi(i-1)}{12}), \sin(\frac{2\pi(i-1)}{12}), 0)$ ($i = 1–4$) and $\mathbf{c}^* = (0,0,c^*)$, with $a^*$ and $c^*$ being $0.6942\,\text{Å}^{-1}$ and $0.6047\,\text{Å}^{-1}$, respectively (Supplementary Fig. 2 and Supplementary Table 1). Complete data on the powder XRD profile calculated for the Ta$_{1.6}$Te dodecagonal QC phase are provided in Supplementary Data 2.

## Electrical resistivity
The temperature dependence of the electrical resistivity normalized by the value at $T = 300$ K, $\rho/\rho_{300\,\mathrm{K}}$ (where $\rho_{300\,\mathrm{K}}$ is approximately $1.7\,\mathrm{m\Omega\,cm}$), is shown in Fig. 2a. The normalized electrical resistivity $\rho/\rho_{300\,\mathrm{K}}$ gradually increased with decreasing temperature and abruptly descended to zero at a low temperature, indicating the emergence of superconductivity. Analysis of the temperature dependence of

$\rho/\rho_{300\,\mathrm{K}}$ below 2 K (Fig. 2b) indicated that $\rho(T)$ abruptly dropped to zero at a midpoint transition temperature $T_c^{\mathrm{mid}} = 0.98$ K, where $\rho(T_c^{\mathrm{mid}}) = 0.5 \times \rho(1.1\,\mathrm{K})$. Examination of the magnetic field dependence of $\rho/\rho_{300\,\mathrm{K}}$ at different temperatures (Fig. 2c) suggests that the superconducting transitions were suppressed by increasing the magnetic field. Applying the criterion of 50% of the normal state resistivity to define the upper critical field $H_{c2}$, we evaluated $H_{c2}$ and plotted the data against temperature (inset in Fig. 2d). This relation was analyzed (Fig. 2d) by plotting $-H_{c2}/\left[T_c\left(\frac{dH_{c2}}{dT}\right)_{T=T_c}\right]$ against $T/T_c$ (that is, $h^*$ vs. $t$). According to the Werthamer–Helfand–Hohenberg theory[20], which considers spin–orbit scattering and spin paramagnetism, $H_{c2}$ in the dirty limit (mean free path $l \ll$ coherence length $\xi_0$) is expressed in terms of the digamma function as:

$$
\ln\frac{1}{t} = \left(\frac{1}{2} + \frac{i\lambda_{SO}}{4\gamma}\right)\psi\left(\frac{1}{2} + \frac{\bar{h} + \frac{1}{2}\lambda_{SO} + i\gamma}{2t}\right) \\
+ \left(\frac{1}{2} - \frac{i\lambda_{SO}}{4\gamma}\right)\psi\left(\frac{1}{2} + \frac{\bar{h} + \frac{1}{2}\lambda_{SO} - i\gamma}{2t}\right) - \psi\left(\frac{1}{2}\right),
\tag{1}
$$

where $\psi$ is the digamma function, $\gamma \equiv \left[(\alpha\bar{h})^2 - (\frac{1}{2}\lambda_{SO})^2\right]^{\frac{1}{2}}$, and $\bar{h} = \frac{4}{\pi^2}h^* = \frac{4H_{c2}}{\pi^2(-dH_{c2}/dt)_{t=1}}$, with parameters $\lambda_{SO}$ and $\alpha$ representing the effects of the spin–orbit scattering and spin paramagnetism, respectively. The obtained data were adequately fit to Eq. (1) using $\alpha = \lambda_{SO} = 0$, indicating that the two effects were negligible. Through this fitting, the value of the zero-temperature upper critical field $H_{c2}(0)$ was deduced to be 32 kOe. The dirty-limit superconductivity was consistent with the large residual resistivity shown in Fig. 2a. A similar $H_{c2}$ tendency has also been observed for the Al–Zn–Mg i-QC[19].

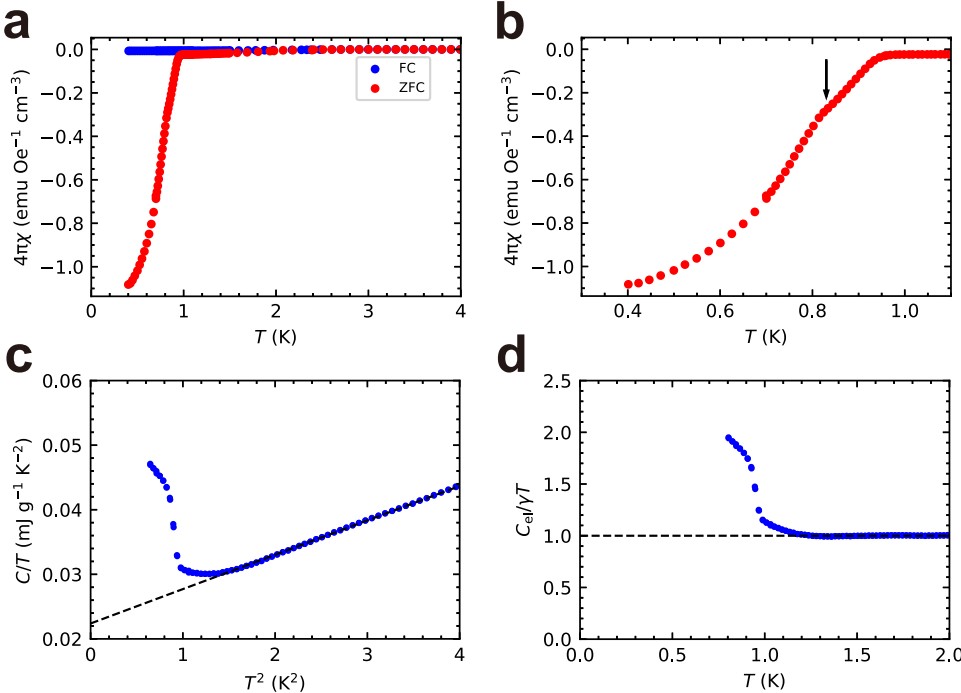

**Fig. 3 | Magnetic susceptibility and specific heat measurements. a** Temperature dependence of the magnetic susceptibility $\chi$. **b** The same as **a**, but in a low temperature region. **c** Temperature dependence of specific heat $C$, analyzed by plotting $C/T$ against $T^2$ and **d**, the ratio between the electronic component of the specific heat $C_{el}$ and $\gamma T$.

## Magnetic susceptibility and specific heat

The magnetic susceptibility and specific heat were investigated to corroborate the occurrence of bulk superconductivity. The temperature dependence of the magnetic susceptibility $\chi$ was monitored at an external magnetic field of 50 Oe under zero-field cooling (ZFC) and field cooling (FC) conditions (Fig. 3a). In the ZFC curve, a sharp drop occurred at an onset temperature of $\approx 0.95$ K—a value consistent with the transition temperature corresponding to $\rho(T)$ in Fig. 2b; moreover, a large diamagnetism was observed, indicating the exclusion of the magnetic flux owing to the shielding effect. The ZFC $4\pi\chi$ reached $-1.1$ emu Oe$^{-1}$ cm$^{-3}$ at the lowest temperature (Fig. 3a), exceeding 100 % diamagnetization. This is because demagnetization-factor correction was not made due to the irregular shape of the sample, leading to an overestimation of $4\pi\chi$. A rough estimation of the demagnetization factor $N$ gave $N \approx 0.25$ (see Methods section). Then, the corrected $4\pi\chi$ at the lowest temperature was calculated to be $-0.86$ emu Oe$^{-1}$ cm$^{-3}$. The ZFC data in a low temperature region is replotted in Fig. 3b, where a slope change is clearly evident at $T = 0.83$ K (see the temperature derivative curve in Supplementary Fig. 11b). This can be attributed to a superconducting transition of an impurity phase, which is most likely a high-order CA phase such as the previously reported CA phase of Ta$_{181}$Te$_{112}$ (refs. 8,9), which has a structure very similar to the dodecagonal QC phase. In any case, our ZFC $4\pi\chi$ data clearly indicates that the superconducting transition at $T = 0.95$ K emanates from the majority phase of the sample, which is the dodecagonal QC phase.

The temperature dependence of the specific heat $C$ was probed by plotting $C/T$ against $T^2$ (Fig. 3c). A jump in the $C$ value was clearly observed at $T \approx 1$ K—a value consistent with the transition temperatures corresponding to $\rho(T)$ (Fig. 2b) and $\chi(T)$ (Fig. 3a). A linear behavior of $C/T$ with $T^2$ of the form $C/T = \gamma + \beta T^2$ was observed at $T^2$ values above ~1.4 K$^2$. Accordingly, the corresponding $\gamma$ and $\beta$ values were estimated as 0.0224 mJ g$^{-1}$ K$^{-2}$ and 0.00532 mJ g$^{-1}$ K$^{-4}$, respectively. The electronic part of the specific heat $C_{el}$ was

determined by subtracting the lattice contribution of the specific heat $\beta T^3$ from the measured specific heat, and $C_{el}/\gamma T$ was subsequently plotted against $T$ (Fig. 3d). The jump height $\Delta C_{el}$ at $T_c$ was $\Delta C_{el} \approx 1.0 \gamma T_c$, which was large enough to confirm the occurrence of bulk superconductivity.

With respect to the jump height $\Delta C_{el}$, the Bardeen–Cooper–Schrieffer (BCS) theory predicts $\Delta C_{el} = 1.43 \gamma T_c$. Recently, Takemori et al[21]. numerically studied the physical properties of quasiperiodic superconductors using an attractive Hubbard model on the Penrose tiling and showed $\Delta C_{el} = 1.21 \gamma T_c$. They demonstrated that this reduction in $\Delta C_{el}$ is in agreement with the experimental results of the Al–Zn–Mg i-QC[19]. The specific heat data of the dodecagonal QC were compared with those of the Al–Zn–Mg i-QC and the theoretical curves (Fig. 4). The data of the dodecagonal QC also shows an obvious reduction in $\Delta C_{el}$ from the BCS value and are in rough agreement with those of the Al–Zn–Mg i-QC. Data at lower temperatures are necessary for more detailed comparisons.

Using the aforementioned value of $\beta$, the Debye temperature $\Theta_D$ was deduced to be 132 K. Furthermore, the electron–phonon coupling constant $\lambda_{ep}$ was obtained using the McMillan equation:[22]

$$T_c = \frac{\Theta_D}{1.45} \exp\left\{ -\frac{1.04(1+\lambda_{ep})}{\lambda_{ep} - \mu^*(1+0.62\lambda_{ep})} \right\}, \quad (2)$$

where $\mu^*$ denotes the Coulomb pseudopotential parameter, which is typically considered 0.13 for superconductors including transition metals. Using $\mu^* = 0.13$, $T_c = 0.98$ K, and $\Theta_D = 132$ K, $\lambda_{ep}$ was calculated to be 0.52, indicating a weak-coupling superconductivity. In general, the density of a QC phase is approximately the same as that of its CA phase. Therefore, the density of the Ta$_{1.6}$Te dodecagonal QC phase was estimated to be 10.6 g cm$^{-3}$, which was calculated for the Ta$_{97}$Te$_{60}$ CA phase from its structure data[10]. Using this value and the estimated values of $\gamma$ and $\lambda_{ep}$, the density of states (for one of the two spin states) at the Fermi level $D(E_F)$ was estimated as $1.24 \times 10^{47}$ states J$^{-1}$ m$^{-3}$)

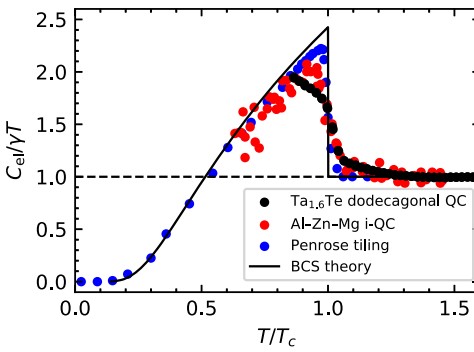

**Fig. 4 | Temperature dependences of specific heats.** The experimental data of the dodecagonal QC (black dots) and the Al–Zn–Mg i-QC (red dots), and the data numerically obtained for the Penrose tiling (blue dots) and the curve predicted by the BCS theory (black line). The data for the Al–Zn–Mg i-QC and Penrose tiling were extracted from the figures in the original papers[19,21].

using the expression[22]:

$$D(E_F) = \frac{3\gamma}{2\pi^2 k_B^2 (1+\lambda_{ep})},$$ (3)

where $k_B$ is the Boltzmann constant.

## Discussion

Superconductivity in vdW layered crystals of tantalum dichalcogenides (TaCh₂; Ch=S, Se, and Te) has been extensively studied in recent years[23–32]. Superconductivity in this group of materials often occurs in competition with a charge density wave (CDW) state. In TaTe₂, a CDW state is stable down to the lowest temperature at ambient pressure without transitioning to a superconducting state; superconductivity appears only when the CDW state is suppressed, e.g., by applied pressure[23]. In $2H$-TaS₂, $4H_b$-TaS₂ and $2H$-TaSe₂, superconductivity develops in the temperature range below that of the CDW state at ambient pressure. For $4H_b$-TaS₂, on the other hand, the realization of chiral superconductivity has recently been suggested[24].

The two-dimensionality of superconductivity is one of the major issues in these materials. The anisotropy parameter $\gamma$ defined as $\gamma = H_{c2}^{\parallel}(0)/H_{c2}^{\perp}(0)$ is approximately 4.5, 17, and 2.9 for $2H$-TaS₂, $4H_b$-TaS₂, and $2H$-TaSe₂, respectively[25–27]. In $4H_b$-TaS₂, the degree of two-dimensionality is considerably high. This is due to the alternating stacking structure of superconducting $1H$ and Mott insulating $1T$ layers. In $2H$-TaS₂ and $2H$-TaSe₂, the anisotropy is not so strong and the superconductivity is essentially three-dimensional with weak anisotropy. However, the degree of two-dimensionality can be further enhanced by weakening the interlayer coupling, for instance, through intercalation. Quasi-two-dimensional superconductivity has been realized in $2H$-TaS₂ samples intercalated with Pb[28] and pyridine[29]. Another direct method to realize two-dimensional superconductivity is to synthesize a very thin sample, for example, by exfoliation. Atomically thin $2H$-TaS₂ and $2H$-TaSe₂ have been shown to exhibit significantly higher $T_c$ than the corresponding bulk values[30,31]. In the absence of in-plane inversion symmetry, Zeeman-type spin–orbit interaction forces the electron spins to align perpendicular to the plane in the two-dimensional limit. The realization of such an 'Ising superconductivity' has been reported for monolayer $1H$-TaS₂[32]. In this context, the elucidation of the two-dimensional nature of superconductivity in the vdW layered quasicrystal of Ta₁.₆Te is an important and challenging task, which will be achieved by the fabrication of sizable single-crystal samples, direction-dependent magnetic and transport measurements, the fabrication of intercalated and exfoliated samples, and their direction-dependent measurements.

**Table 1 | Superconducting characteristics of the Ta₁.₆Te dodecagonal quasicrystal (dd-QC), other Ta–chalcogenide systems, Ta, and the Al–Zn–Mg i-QC. $T_c$ and $\Theta_D$ were sourced from the literature, except for $4H_b$-TaS₂**

|  | $T_c$ (K) | $\Theta_D$ (K) | $\lambda_{ep}$ | $D(E_F)$ ($10^{47}$ states J⁻¹ m⁻³) | References |
|---|---|---|---|---|---|
| Ta₁.₆Te dd-QC | 0.98 | 132 | 0.52 | 1.24 | Present study |
| Ta₂Se | 3.8 | 271 | 0.61 | 1.82 | 33 |
| $2H$-TaSe₂ | 0.15 | 202 | 0.37 | 0.685 | 34 |
| $2H$-TaS₂ | 0.8 | 236 | 0.45 | 1.19 | 34 |
| $4H_b$-TaS₂ | 2.7 | 241 | 0.57 | 0.805 | 24 |
| Ta | 4.4 | 231 | 0.67 | 2.39 | 35 |
| Al–Zn–Mg i-QC | 0.05 | 317 | 0.27 | 0.453 | 19 |

$T_c$ and $\Theta_D$ for $4H_b$-TaS₂ were obtained by fitting the specific heat data presented in the literature. $\lambda_{ep}$ was calculated from $T_c$ and $\Theta_D$ using Eq. (2), where $\mu^*$ values of 0.1 and 0.13 were used for the Al–Zn–Mg i-QC and the other systems, respectively. $D(E_F)$ was calculated from $\lambda_{ep}$ and reported $\gamma$ using Eq. (3)

The superconducting parameters of the Ta–chalcogenides[24,33,34] including Ta-rich Ta₂Se and Ta₁.₆Te dodecagonal QC (from the present study), together with those of Ta[35], and Al–Zn–Mg i-QC[19] are summarized in Table 1. For a simple metal system of the Al–Zn–Mg i-QC, $\mu^* = 0.1$ was used[22] in the calculation of $\lambda_{ep}$, while for the others $\mu^* = 0.13$ was used. The $\lambda_{ep}$ values of Ta and Ta₂Se were fairly high, which may be in an intermediate-coupling range. The other systems exhibited sufficiently small $\lambda_{ep}$ values appropriate for weak-coupling superconductors, with the $\lambda_{ep}$ value of the Al–Zn–Mg i-QC being considerably smaller than those of the others. Also for $D(E_F)$, the Al–Zn–Mg i-QC exhibited the smallest value among the specimens. For layered Ta–chalcogenides, $\Theta_D$ scaled with $M_{ave}^{-1/2}$ ($M_{ave}$: average atomic weight) except for Ta₂Se, as shown in Supplementary Discussion 2 (Supplementary Fig. 10).

Recent theoretical studies[36–38] have revealed that quasicrystalline superconductors exhibit several unconventional behaviors that are typically not observed in other known superconductors in periodic and disordered systems, thus opening a new field in the research of superconductivity. Nagai[36] has studied superconducting tight-binding models of Penrose and Ammann–Beenker lattices (typical two-dimensional quasicrystalline lattices) and demonstrated an intrinsic vortex pinning due to spatially inhomogeneous superconducting order parameter. Such an inhomogeneous order parameter arises from the quasicrystalline structural order, and therefore, the vortex pinning occurs without an impurity or defect. Sakai et al[37]. have investigated quasicrystalline superconductivity using an attractive Hubbard model on a Penrose lattice using the real-space dynamical mean-field theory. Unconventional spatially-extended Cooper pairs were formed; the sum of the momenta of the Cooper pair electrons was nonzero, in contrast to the zero total momentum of the Cooper pair in the conventional BCS superconductivity. Such a nonzero total momentum of the Cooper pair is also observed for the Fulde–Ferrell–Larkin–Ovchinnikov (FFLO) state previously proposed for periodic systems[39–42]. However, the unconventional Cooper pairing in the model QC is completely different from the FFLO state because the Cooper pairing occurs under no magnetic field. In addition, under a high magnetic field, a state similar to the FFLO state is formed in the model QC[38]. However, this state is also different from the conventional FFLO state in periodic systems and forms a fractal-like spatial pattern of the oscillating superconducting order parameter, which is compatible with the self-similar structural order that is possessed by the QCs. As mentioned above, many interesting features are theoretically expected for superconducting QCs, which are yet to be demonstrated experimentally, and the Ta₁.₆Te dodecagonal QC phase in the present study offers a precious platform for it.

In conclusion, polygrain $Ta_{1.6}Te$ dodecagonal QC samples were fabricated by reaction sintering. Careful phase identification of the sample was performed by electron and powder X-ray diffraction experiments and diffraction-profile simulations. The samples were subjected to electrical resistivity, magnetic susceptibility, and specific heat measurements. The results unconditionally validate the occurrence of bulk superconductivity at a $T_c$ of ~1 K. This is the first example of superconductivity in thermodynamically stable QCs. These findings are expected to motivate further investigations into the physical properties of vdW layered quasicrystals as well as two-dimensional quasicrystals. In particular, the dodecagonal QC provides a valuable platform for the experimental demonstration of the unique superconductivity theoretically predicted for QCs.

## Methods

### Sample synthesis

$Ta_{1.6}Te$ samples were synthesized by reaction sintering. $TaTe_2$ and Ta were used as the starting materials at a ratio of 1:3. $TaTe_2$ was prepared in advance from a 1:2 elemental powder mixture of Ta (−325 mesh, 99.9%; Rare Metallic) and Te (−100 mesh, 99.99%; Rare Metallic). The mixture was pressed into a pellet (diameter of 10 mm) and inserted into a quartz tube, which was evacuated to $5-6 \times 10^{-4}$ Pa, sealed, and then subjected to two-step annealing (773 K for 24 h, followed by 1273 K for 24 h). Subsequently, the mixture of the $TaTe_2$ powder, Ta powder, and a shot (30–40 mg) of iodine (99.9%; FUJIFILM Wako Pure Chemical; used to promote the reaction) was pressed into a 5-mm-thick pellet (diameter of 10 mm). The pellet was sealed in an evacuated ($5-6 \times 10^{-4}$ Pa) quartz tube and sintered at 1273 K for six days.

### Sample characterization

The structures of the sintered samples were examined by powder X-ray diffractometry (XRD) and electron diffractometry. Powder XRD profiles were obtained using a diffractometer (RINT-2500V; Rigaku) with Cu Kα radiation at 40 kV and 200 mA, a divergence slit open-angle of 0.5°, scattering slit open-angle of 0.5°, receiving slit width of 0.15 mm, rate of 0.5° min$^{-1}$, and step of 0.01°. Electron diffraction patterns were obtained using a transmission electron microscope (JEM-2010F; JEOL) operated at an acceleration voltage of 200 kV.

### Physical property measurements

The electrical resistivity was measured using a physical property measurement system (PPMS; Quantum Design) with a four-probe method and a direct current of 25–100 μA. Samples with cross-sectional areas of 2–3 mm × 2–3 mm and lengths of 5–6 mm were used for the electrical resistivity measurements. The magnetic field was applied in the direction roughly perpendicular to the electrical current. The sample for magnetic susceptibility measurements was prepared by crushing the sintered pellet. A piece of 95.64 mg was subjected to the magnetic susceptibility measurements using a magnetic property measurement system (MPMS3; Quantum Design) equipped with a $^3$He refrigerator. Using the density (10.6 g cm$^{-3}$) calculated for the CA phase[10], which should be approximately the same as the density of the dodecagonal QC phase, the volume was estimated to be 9.02 mm³. The ratio of the typical length of the sample along z-direction to that in xy-plane (z//**H**) was roughly 1.5, giving the demagnetization factor of $N \approx 0.25$ (ref. [43]). The sample for specific heat measurements was prepared also by crushing the sintered pellet. A piece was picked up and roughly shaped and polished with sandpaper so that the side to be placed on the sample platform is flat. A sample with a weight of 26.09 mg and the size of $\sim 2.2 \times 2.2 \times 0.5$ mm³ was subjected to the specific heat measurements using the PPMS with the thermal relaxation method.

## Data availability

The data that support the findings of this study are available within the paper and its Supplementary Information. Source data are provided with this paper.

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

## Acknowledgements

This study was supported by the JST-CREST program (grant no. JPMJCR22O3; Japan) (R.T. and K.E.) and JSPS KAKENHI Grant Numbers JP19H05821 (R.T. and K.E.) and JP23K04355 (Y.T.). A part of this work was performed using facilities of the Cryogenic Research Center, the University of Tokyo, and Advanced Research Infrastructure for Materials and Nanotechnology in Japan (ARIM) of the Ministry of Education, Culture, Sports, Science and Technology (MEXT), grant JPMXP1222UT0053.

## Author contributions

K.H., S.N., and Y.T. fabricated the samples and performed powder XRD measurements. S.N., Y.K. and K.E. analyzed the powder XRD data and constructed the profile. Y.K., K.H., and Y.T. performed the electron dif-fraction experiments. Y.T., K.H., and K.E. conducted the electrical resistivity and specific heat measurements. S.S., R.T., Y.T., and K.H. conducted the magnetic susceptibility experiments. K.E. designed and supervised the whole project. Y.T. and K.E. cowrote the manuscript. All the authors have discussed the results and contributed to the manuscript.

## Competing interests

The authors declare no competing interests.
