## [Peer Review File · Nature Communications]

Superconductivity in a van der Waals layered quasicrystalREVIEWER COMMENTS

Reviewer #1 (Remarks to the Author):

The paper presents a very interesting observation of a superconducting state in a dodecagonal quasicrystal in the TaTe systems. This system is a layered structure with Van der Waals interaction between layers. Evidence of sample quasicrystallinity are given by electron diffraction (the powder data are less convincing, see hereafter). Resistivity, susceptibility and specific heat measurements all point towards a bulk superconductivity with a T_c of 1K. The analysis of the data indicate a BCS type superconductivity although further low T C_p measurement are necessary to confirm this point.

The results are very interesting and of broad interest and should be published.

However the authors need to improve their manuscript especially the introduction and conclusion and answer to the following points.

The introduction with the explanation of what is a quasicrystal should be recast. Although it is important to cite the original papers, the notion of quasicrystals should be placed in the today's knowledge and context and in particular the link with Aperiodic Crystals should be made. The definition should refer to the IUCr one or text books.

The conclusion needs to be completely rewritten. As it is, the manuscript ends rather abruptly after a discussion and a small paragraph. The conclusion should gather most important findings and eventual perspectives.

The discussion on the nature of the superconductivity is interesting. In the case of the AlZnMg one, it seems that the BCS theory modified for quasicrystal explains the specific heat measurement. Can this be the case also for this QC? What does the 2D character brings about in this case? Is a strong anisotropy expected? What can be said also with the electronic calculation of ref 6. All those points should be included in the discussion.

The Debye temperature (130K) is extremely small, and almost half that of TaSe₂. This is rather strange and some arguments should be given to explain this exceptionally low value, or the extraction of the Debye temperature rechecked.

Do not use the abbreviation dd for dodecagonal: use either 12-fold or dodecagonal.

XRD characterisation

To be convincing all XRD plots should be made with a logarithmic or $\sqrt{2}$ intensity scale ; otherwise details of weak peaks are invisible.

Calculation of dodecagonal QC diffraction pattern is puzzling: in Fig 7 b supplementary, the 12-fold symmetry should be observed whereas this is not the case. In fact one needs to average the intensities of the Crystal Approximant so that the 12fold symmetry is obtained . It seems that this averaging has not been performed.

Reviewer #2 (Remarks to the Author):

Superconductivity in quasicrystalline materials is a very interesting topic. In this manuscript, the authors present synthesis and physical property characterization results of the only van der Waals layered quasicrystal Ta_{1.6}Te. Superconductivity is reported to exist at ~ 1 K.

The reviewer finds the evidence for superconductivity is solid, although multiple superconducting phases may exist, as observed transition shows step like features. This phase purity issue can be quite important. Earlier report on Ta_{1.6}Te indicates that using the synthesis condition similar to the one used in current manuscript will result in a mixed quasicrystal and approximant phases. Electron diffraction from certain areas of the sample clearly show periodic superstructure that's related to the approximant phase (PNAS 117, 26135). In the current manuscript, although electron diffraction is shown, it is unclear if the information on spatial variation of phases is captured. In addition, the powder xray diffraction has a low signal to noise ratio. This data quality is reasonable for the quasicrystalline system under study. But it does not offer clean evidence for sample purity. In fact, one can suspect that according to Figure S9, the observed diffraction peak profile for 10001, 11100 and 11101 may be consistent with the production of approximant phases. Therefore, the reviewer is not fully convinced of the titled claim.

Response to comments

We would like to thank the reviewers for taking their time to review our manuscript and for their valuable comments for improving our paper. Below, we respond to each comment one by one and explain how we have revised the manuscript. All changes in the revised manuscript are highlighted in red.

Reviewer #1

Comment (1)

The introduction with the explanation of what is a quasicrystal should be recast. Although it is important to cite the original papers, the notion of quasicrystals should be placed in the today's knowledge and context and in particular the link with Aperiodic Crystals should be made. The definition should refer to the IUCr one or text books.

Response (1)

We have revised the introduction in accordance with the reviewer's suggestion. We added:

'In 1992, the International Union of Crystallography modified the definition of crystals in order to incorporate QCs into crystals⁵, and revised the classification scheme and definition of QCs accordingly. Now, all solids with long-range ordered atomic arrangements are called crystals. This includes 'aperiodic crystals', whose long-range order is not periodic but quasiperiodic. QCs are defined as aperiodic crystals that are neither incommensurate modulated structures nor aperiodic composite crystals. One significant difference between QCs and periodic crystals is that QCs are allowed to have a crystallographically forbidden rotational symmetry, which is, though, not a necessary but a sufficient condition for a QC in today's context.'

(revised manuscript: page 3, line 7—)

Comment (2)

The conclusion needs to be completely rewritten. As it is, the manuscript ends rather abruptly after a discussion and a small paragraph. The conclusion should gather most important findings and eventual perspectives.

Response (2)

We modified conclusion as follows.

‘In conclusion, polygrain Ta_{1.6}Te dodecagonal QC samples were fabricated by reaction sintering. Careful phase identification of the sample was performed by electron and powder X-ray diffraction experiments and diffraction-profile simulations. The samples were subjected to electrical resistivity, magnetic susceptibility, and specific heat measurements. The results unconditionally validate the occurrence of bulk superconductivity at a T_c of ~ 1 K. This is the first example of superconductivity in thermodynamically stable QCs. These findings are expected to motivate further investigations into the physical properties of vdW layered quasicrystals as well as two-dimensional quasicrystals. In particular, the dodecagonal QC provides a valuable platform for the experimental demonstration of the unique superconductivity theoretically predicted for QCs.’

(revised manuscript: page 12, line 16—)

Comment (3)

The discussion on the nature of the superconductivity is interesting. In the case of the AlZnMg one, it seems that the BCS theory modified for quasicrystal explains the specific heat measurement. Can this be the case also for this QC? What does the 2D character brings about in this case? Is a strong anisotropy expected? What can be said also with the electronic calculation of ref 6. All those points should be included in the discussion.

Response (3)

Specific heat

In response to the comment on the specific heat, we added a figure that compares our data with those of the Al–Zn–Mg i-QC, the BCS curve, and Takemori’s calculations (PRB, 102, 115108). Here, we assume that the reviewer refers to Takemori’s calculations by mentioning ‘BCS theory modified for quasicrystal’. We added the following paragraph:

‘With respect to the jump height ΔC_{el} , the Bardeen–Cooper–Schrieffer (BCS) theory predicts $\Delta C_{el} = 1.43 \gamma T_c$. Recently, Takemori et al.²¹ numerically studied the physical properties of quasiperiodic superconductors using an attractive Hubbard model on the Penrose tiling and showed $\Delta C_{el} = 1.21 \gamma T_c$. They demonstrated that this reduction in ΔC_{el} is in agreement with the experimental results of the Al–Zn–Mg i-QC¹⁹. The specific heat data of the dodecagonal QC were compared with those of the Al–Zn–Mg i-QC and the theoretical curves (Fig. 4). The data of the dodecagonal QC also shows an obvious reduction in ΔC_{el} from the BCS value and are in rough agreement with those of the Al–Zn–Mg i-QC. Data at lower temperatures are necessary for more detailed comparisons.’

(revised manuscript: page 8, line 16—)

and we added Fig. 4:

Fig. 4 | Temperature dependences of specific heats. The experimental data of the dodecagonal QC (black dots) and the Al–Zn–Mg i-QC (red dots), and the numerically obtained data for the Penrose tiling (blue dots) and the curve predicted by the BCS theory (black line). The data for the Al–Zn–Mg i-QC and the Penrose tiling have been reproduced from the original papers^{19,21}.

2D character / anisotropy

Yes, we can expect an anisotropy in the dodecagonal QC based on recent works on related systems. Moreover, we can even control the degree of 2D character as described below. Currently, we have no data on the 2D character or anisotropy since sizable single-crystal samples are required to perform direction-dependent magnetic and transport measurements. In response to the reviewer’s comment, however, we added a brief review of superconductivity in vdW layered crystals of tantalum dichalcogenides (TaCh₂; Ch=S, Se, and Te), focusing on its two-dimensional nature, and, based on this, presented a couple of methods to enhance the 2D character in the dodecagonal QC.

We added the following two paragraphs:

‘Superconductivity in vdW layered crystals of tantalum dichalcogenides (TaCh₂; Ch=S, Se, and Te) has been extensively studied in recent years^{23–32}. Superconductivity in this group of materials often occurs in competition with a charge density wave (CDW) state. In TaTe₂, a CDW state is stable down to the lowest temperature at ambient pressure without transitioning to a superconducting state; superconductivity appears only when the CDW state is suppressed, e.g., by applied pressure²³. In 2H-TaS₂, 4H_b-TaS₂ and 2H-TaSe₂, superconductivity develops in the temperature range below that of the CDW state at ambient pressure. For 4H_b-TaS₂, on the other hand, the realization of chiral superconductivity has recently been suggested²⁴.

The two-dimensionality of superconductivity is one of the major issues in these materials. The anisotropy parameter γ defined as $\gamma = H_{c2}^{\parallel}(0)/H_{c2}^{\perp}(0)$ is approximately 4.5, 17, and 2.8 for 2H-TaS₂, 4H_b-TaS₂, and 2H-TaSe₂, respectively^{25–27}. In 4H_b-TaS₂, the degree of two-dimensionality is considerably high. This is due to the alternating

stacking structure of superconducting $1H$ and Mott insulating $1T$ layers. In $2H$ -TaS₂ and $2H$ -TaSe₂, the anisotropy is not so strong and the superconductivity is essentially three-dimensional with weak anisotropy. However, the degree of two-dimensionality can be further enhanced by weakening the interlayer coupling, for instance, through intercalation. Quasi-two-dimensional superconductivity has been realized in $2H$ -TaS₂ samples intercalated with Pb²⁸ and pyridine²⁹. Another direct method to realize two-dimensional superconductivity is to synthesize a very thin sample, for example, by exfoliation. Atomically thin $2H$ -TaS₂ and $2H$ -TaSe₂ have been shown to exhibit significantly higher T_c than the corresponding bulk values^{30,31}. In the absence of in-plane inversion symmetry, Zeeman-type spin-orbit interaction forces the electron spins to align perpendicular to the plane in the two-dimensional limit. The realization of such an ‘Ising superconductivity’ has been reported for monolayer $1H$ -TaS₂³². In this context, the elucidation of the two-dimensional nature of superconductivity in the vdW layered quasicrystal of Ta_{1.6}Te is an important and challenging task, which will be achieved by the fabrication of sizable single-crystal samples, direction-dependent magnetic and transport measurements, the fabrication of intercalated and exfoliated samples, and their direction-dependent measurements.’

(revised manuscript: page 9, line 18—)

Electronic calculations in (PNAS 117, 26135)

It seems impossible to extract meaningful information about anisotropy, the nature of the bands contributing to conduction (Ta- d , Te- p or what?), DOS(E), etc. from the very limited information on the E - k relations (only along the symmetry directions in the Brillouin zone) presented in the paper. Therefore, discussion based on this work is not included in the revised manuscript.

Comment (4)

The Debye temperature (130K) is extremely small, and almost half that of TaSe₂. This is rather strange and some arguments should be given to explain this exceptionally low value, or the extraction of the Debye temperature rechecked.

Response (4)

In the figure attached at the end of this response, the Debye temperatures (θ_D) of various layered Ta-chalcogenides, including those in Table 1, are plotted against $M_{\text{ave}}^{-1/2}$ (M_{ave} : average atomic weight). The relationship $\theta_D \propto M_{\text{ave}}^{-1/2}$ approximately holds except for Ta₂Se, showing that the value obtained for the dodecagonal QC is not exceptionally small but is understood by its small value of $M_{\text{ave}}^{-1/2}$. What is exceptional is the large value of Ta₂Se reported by Gui et al., although the reason of which is not clear. We mentioned this point in the text and added the figure to the Supplementary information as below.

Supplementary Discussion 2. Debye temperatures

In Supplementary Fig. 10, the Debye temperatures (θ_D) of various layered Ta-chalcogenides⁶⁻¹⁰, including those in Table 1, are plotted against $M_{\text{ave}}^{-1/2}$ (M_{ave} : average atomic weight). The relationship $\theta_D \propto M_{\text{ave}}^{-1/2}$ approximately holds except for Ta₂Se, whose θ_D is considerably high for unknown reasons.

(revised Supplementary information: page 8, line 22—)

References

6. Gui, X., Górnicka, K., Chen, Q., Zhou, H., Klimczuk, T. & Xie, W. Superconductivity in metal-rich chalcogenide Ta₂Se. *Inorg. Chem.* **59**, 5798–5802 (2020).
7. Bhoi, D., Khim, S., Nam, W., Lee, B. S., Kim, C., Jeon, B.-G., Min, B. H., Park, S. & Kim, K. H. Interplay of charge density wave and multiband superconductivity in 2H-Pd_xTaSe₂. *Sci. Rep.* **6**, 24068 (2016).
8. Luo, H., Xie, W., Tao, J., Inoue, H., Gyenis, A., Krizan, J. W., Yazdani, A., Zhu, Y. & Cava, R. J. Polytypism, polymorphism, and superconductivity in TaSe_{2-x}Te_x. *Proc. Natl. Acad. Sci. U.S.A.* **112**, E1174–E1180 (2015).
9. Wagner, K. E., Morosan, E., Hor, Y. S., Tao, J., Zhu, Y., Sanders, T., McQueen, T. M., Zandbergen, H. W., Williams, A. J., West, D. V. & Cava, R. J. Tuning the charge

density wave and superconductivity in Cu_xTaS_2 . *Phys. Rev. B* **78**, 104520 (2008).

10. Ribak, A., Skiff, R. M., Mograbi, M., Rout, P. K., Fischer, M. H., Ruhman, J., Chashka, K., Dagan, Y. & Kanigel, A. Chiral superconductivity in the alternate stacking compound 4Hb-TaS_2 . *Sci. Adv.* **6**, eaax9480 (2020).

(revised Supplementary information: page 9, line 15—)

Supplementary Fig. 10 | Debye temperatures (θ_D) plotted against $M_{\text{ave}}^{-1/2}$ (M_{ave} : average atomic weight).

(revised Supplementary information: page 27)

We also added the following sentence in main text:

‘For layered Ta–chalcogenides, θ_D scaled with $M_{\text{ave}}^{-1/2}$ (M_{ave} : average atomic weight) except for Ta_2Se , as shown in Supplementary Discussion 2 (Supplementary Fig. 10).’

(revised manuscript: page 11, line 11—)

Comment (5)

Do not use the abbreviation dd for dodecagonal: use either 12-fold or dodecagonal.

Response (5)

We replaced ‘dd’ with ‘dodecagonal’ throughout the text, as requested by the reviewer.

Comment (6)

XRD characterisation

To be convincing all XRD plots should be made with a logarithmic or sqrt(2) intensity scale ; otherwise details of weak peaks are unvisble.

Response (6)

The XRD profiles in Fig. 1b and Supplementary Figs. 4 and 8 were plotted on a logarithmic scale as requested by the reviewer, whereas the XRD patterns in Fig. 1c and Supplementary Figs. 2b and 9 are left unchanged since these plots are already magnified and weak peaks are clearly visible. The Supplementary Fig. 1 is also unchanged since the TaTe₂ and Ta data, taken from the 'Powder Diffraction Datafile (PDF)', are better displayed in a linear scale.

Comment (7)

Calculation of dodecagonal QC diffraction pattern is puzzling: in Fig 7 b supplementary, the 12-fold symmetry should be observed whereas this is not the case. In fact one needs to average the intensities of the Crystal Approximant so that the 12fold symmetry is obtained . It seems that this averaging has not been performed.

Response (7)

We performed the averaging in Supplementary Fig. 7, as requested by the reviewer.

Reviewer #2

Comment (8)

The reviewer finds the evidence for superconductivity is solid, although multiple superconducting phases may exist, as observed transition shows step like features. This phase purity issue can be quite important. Earlier report on Ta_{1.6}Te indicates that using

the synthesis condition similar to the one used in current manuscript will result in a mixed quasicrystal and approximant phases. Electron diffraction from certain areas of the sample clearly show periodic superstructure that's related to the approximant phase (PNAS 117, 26135). In the current manuscript, although electron diffraction is shown, it is unclear if the information on spatial variation of phases is captured. In addition, the powder xray diffraction has a low signal to noise ratio. This data quality is reasonable for the quasicrystalline system under study. But it does not offer clean evidence for sample purity. In fact, one can suspect that according to Figure S9, the observed diffraction peak profile for 10001, 11100 and 11101 may be consistent with the production of approximant phases. Therefore, the reviewer is not fully convinced of the titled claim.

Response (8)

First, (1) the ZFC magnetic susceptibility and specific heat measurements showed no transition other than the one at 1K in the measured temperature range, which rules out the possibility of multiple superconducting phases. Second, (2) the acquired powder XRD profile is consistent with the simulated profile of the dodecagonal QC, but apparently different from those of the CA phases of $Ta_{21}Te_{13}$ and $Ta_{97}Te_{60}$ (see, in particular, the 11100 and 11101 peaks in Supplementary Fig. 9). Third, since powder XRD is less powerful than electron diffraction in distinguishing higher-order CA phases such as $Ta_{181}Te_{112}$ reported in (PNAS 117, 26135), we thoroughly investigated the present sample by electron diffraction, but (3) we did not detect the periodic superstructure of the $Ta_{181}Te_{112}$ CA phase in any of the domains investigated. Thus, the results (1)–(3) lead us to the conclusion that at least **the majority phase is the dodecagonal QC**. However, due to the rather poor data quality in the XRD profile, and also due to the fact that electron diffraction is essentially a local probe and in principle cannot acquire the bulk information of the sample, we agree with the reviewer that possible contamination of the higher-order CA phases cannot be completely ruled out. Therefore, we removed the expression ‘single-phase’ (page 4, line 19, in the original manuscript) and the sentence ‘the sample comprised entirely of the $Ta_{1.6}Te$ dd-QC phase’

(page 5, line 17, in the original manuscript). However, the magnitude of the diamagnetism ($4\pi\chi \approx -1 \text{ emu}/(\text{Oe}\cdot\text{cm}^3)$; Fig. 3a) and the specific heat jump height ($\Delta C_{\text{el}} \approx 1.0 \gamma T_c$; Figs. 3c and 4) clearly demonstrates that **the 1K superconductivity is due to the majority phase.**

From the above two reasons shown in red, we are directly led to the conclusion that the dodecagonal QC phase is responsible for the 1K superconductivity.

Other changes

1. The scale bars in Fig. 1a and Supplementary Fig. 2a were modified.
2. The data of $4H_b\text{-TaS}_2$ were added to Table 1.
3. The first paragraph in the section ‘Discussion and outlook’ in the original manuscript was moved to the section ‘Magnetic susceptibility and specific heat’.
4. References 5 and 23–32 were added, and the old references were renumbered accordingly.
5. We added the following sentence concerning the relationship between the direction of the magnetic field and the electrical current in the Method section:

‘The magnetic field was applied in the direction roughly perpendicular to the electrical current.’

(revised manuscript: page 14, line 6—)

6. We modified the section headings and order according to the Nature Communications formatting instructions.
7. We modified the expression of inverse unit dimensions throughout the manuscript according to the Nature Communications formatting instructions.
8. We modified the Data availability section.

REVIEWER COMMENTS

Reviewer #1 (Remarks to the Author):

In the revised manuscript the authors have answered all the remarks raised in the referee reports. The manuscript can thus be published in its present form.

Reviewer #2 (Remarks to the Author):

The authors' response largely addressed my previous comments. Particularly, I agree with the authors that the superconducting transition reaches to a bulk level, judging from the specific heat jump. As the majority of the sample can be associated with the dodecagonal quasicrystal, the quasicrystalline phase is likely superconducting. Although heat capacity is a better measure, since the authors showed magnetization data, it would be nice to add details in the methods section on the shape of the magnetic susceptibility measurement sample, and corresponding demagnetization factor that was used to plot Fig. 3a.

I respectfully disagree with the authors on the single superconducting transition claim. From Fig. 3a, the ZFC magnetic susceptibility clearly shows a step feature slightly below the onset temperature. If one performs a temperature derivative of the curve, there will likely be two features. Similarly, in Fig. 3b, the superconducting heat capacity jump shows a shoulder at about 1 K and then continues to increase at lower temperatures at a lower rate. This feature is consistent with magnetic susceptibility data. As the authors suggested, maybe some higher order approximant phase still exist in the sample. If so, both phases may be superconducting at relatively close temperatures, which does not go against the claim of the current manuscript.

Response to the comments

Reviewer #1

We thank the reviewer for reviewing our revised manuscript and for his/her recommendation for the publication.

Reviewer #2

We thank the reviewer for reviewing our revised manuscript and for his/her additional valuable comments to further improve our paper. Below, we respond to each comment one by one and explain how we have revised the manuscript. All changes in the revised manuscript are highlighted in red.

Comment (1)

The authors' response largely addressed my previous comments. Particularly, I agree with the authors that the superconducting transition reaches to a bulk level, judging from the specific heat jump. As the majority of the sample can be associated with the dodecagonal quasicrystal, the quasicrystalline phase is likely superconducting.

Response (1)

We are pleased to hear that the previous revision was convincing enough to justify that the quasicrystalline phase is superconducting.

Comment (2)

Although heat capacity is a better measure, since the authors showed magnetization data, it would be nice to add details in the methods section on the shape of the magnetic susceptibility measurement sample, and corresponding demagnetization factor that was used to plot Fig. 3a.

Response (2)

In Fig. 3a, demagnetization-factor correction was not made, because accurate correction was difficult due to the irregular shape of the sample. This is the reason why $4\pi\chi$ reaches $-1.1 \text{ emu Oe}^{-1} \text{ cm}^{-3}$ at the lowest temperature in Fig. 3a, exceeding 100 % diamagnetization. The ratio of the typical length of the sample along z -direction to that in xy -plane ($z//\mathbf{H}$) was roughly 1.5, giving the demagnetization factor of $N \approx 0.25$. Then, the corrected $4\pi\chi$ at the lowest temperature is $4\pi\chi \approx -0.86$. We added the following in main text:

‘The ZFC $4\pi\chi$ reached $-1.1 \text{ emu Oe}^{-1} \text{ cm}^{-3}$ at the lowest temperature (Fig. 3a), exceeding 100 % diamagnetization. This is because demagnetization-factor correction was not made due to the irregular shape of the sample, leading to an overestimation of $4\pi\chi$. A rough estimation of the demagnetization factor N gave $N \approx 0.25$ (see Methods section). Then, the corrected $4\pi\chi$ at the lowest temperature was calculated to be $-0.86 \text{ emu Oe}^{-1} \text{ cm}^{-3}$.’
(2nd revised manuscript: page 7, line 17—)

We added the following in Methods section:

‘The sample for magnetic susceptibility measurements was prepared by crushing the sintered pellet. A piece of 95.64 mg was subjected to the magnetic susceptibility measurements using a magnetic property measurement system (MPMS3; Quantum Design) equipped with a ^3He refrigerator. Using the density (10.6 g cm^{-3}) calculated for the CA phase¹⁰, which should be approximately the same as the density of the dodecagonal QC phase, the volume was estimated to be 9.02 mm^3 . The ratio of the typical length of the sample along z -direction to that in xy -plane ($z//\mathbf{H}$) was roughly 1.5, giving the demagnetization factor of $N \approx 0.25$ (ref. 43).’
(2nd revised manuscript: page 14, line 9—)

We added ref. 43

43. Sato, M. & Ishii, Y. Simple and approximate expressions of demagnetizing factors of

uniformly magnetized rectangular rod and cylinder. *J. Appl. Phys.* **66**, 983–985 (1989).

Comment (3)

I respectfully disagree with the authors on the single superconducting transition claim. From Fig. 3a, the ZFC magnetic susceptibility clearly shows a step feature slightly below the onset temperature. If one performs a temperature derivative of the curve, there will likely be two features. Similarly, in Fig. 3b, the superconducting heat capacity jump shows a shoulder at about 1 K and then continues to increase at lower temperatures at a lower rate. This feature is consistent with magnetic susceptibility data. As the authors suggested, maybe some higher order approximant phase still exist in the sample. If so, both phases may be superconducting at relatively close temperatures, which does not go against the claim of the current manuscript.

Response (3)

We particularly thank the reviewer for this insightful comment. In response to this comment, we reexamined our ZFC data. In the figure attached at the end of this response, the ZFC $4\pi\chi(T)$ curve and its temperature derivative are shown. As the temperature decreases, the derivative curve shows a sharp increase at two points: their onset temperatures are 0.98 and 0.83 K. As the reviewer suggests, the anomaly at 0.83 K possibly corresponds to a superconducting transition of an impurity phase, which is most likely a high-order approximant phase. A sign of such a two-step transition can also be seen in the specific heat data in Fig. 3b, as pointed out by the reviewer. We added the following in main text:

‘The ZFC data in a low temperature region is replotted in Fig. 3b, where a slope change is clearly evident at $T = 0.83$ K (see the temperature derivative curve in Supplementary Fig. 11b). This can be attributed to a superconducting transition of an impurity phase, which is most likely a high-order CA phase such as the previously reported CA phase of $\text{Ta}_{181}\text{Te}_{112}$ (ref. 8 and 9), which has a structure very similar to the dodecagonal QC phase. In any case, our ZFC $4\pi\chi$ data clearly indicates that the superconducting transition at

$T = 0.95$ K emanates from the majority phase of the sample, which is the dodecagonal QC phase.’

(2nd revised manuscript: page 8, line 3—)

We added a new Fig. 3b and moved old Fig. 3b and 3c to new Fig. 3c and 3d, respectively.

Fig. 3 | Magnetic susceptibility and specific heat measurements. a, Temperature dependence of the magnetic susceptibility χ . **b**, The same as **a**, but in a low temperature region. **c**, Temperature dependence of specific heat C , analyzed by plotting C/T against T^2 ; and **d**, the ratio between the electronic component of the specific heat C_{el} and γT .

We also added Supplementary Fig. 11

Supplementary Fig. 11 | ZFC magnetic susceptibility data. **a**, Temperature dependence of the magnetic susceptibility (the same as Fig. 3b); **b**, its temperature derivative. As the temperature decreases, the derivative curve shows a sharp increase at two points: their onset temperatures are 0.98 and 0.83 K.

Other changes

1. We added the description below about the details of the sample for specific heat measurements in Methods section:

‘The sample for specific heat measurements was prepared also by crushing the sintered pellet. A piece was picked up and roughly shaped and polished with sandpaper so that the side to be placed on the sample platform is flat. A sample with a weight of 26.09 mg and the size of $\sim 2.2 \times 2.2 \times 0.5 \text{ mm}^3$ was subjected to the specific heat measurements using the PPMS with the thermal relaxation method.’

(2nd revised manuscript: page 14, line 17—)

REVIEWERS' COMMENTS

Reviewer #2 (Remarks to the Author):

The revised manuscript has properly addressed all my previous comments.

Superconductivity in a van der Waals layered quasicrystal

**Yuki Tokumoto^{1,*}, Kotaro Hamano¹, Sunao Nakagawa¹, Yasushi Kamimura¹,
Shintaro Suzuki², Ryuji Tamura³ & Keiichi Edagawa^{1,*}**

¹*Institute of Industrial Science, The University of Tokyo, Tokyo 153-8505, Japan*

²*Department of Physical Science, Aoyama Gakuin University, Kanagawa, 252-5258,
Japan*

³*Department of Materials Science and Technology, Tokyo University of Science, Tokyo
125-8585, Japan*

* e-mail: tokumoto@iis.u-tokyo.ac.jp; edagawa@iis.u-tokyo.ac.jp

Abstract

Van der Waals layered transition-metal chalcogenides are drawing significant attention owing to their intriguing physical properties. This group of materials consists of abundant members with various elements, having a variety of different structures. However, they are all crystalline materials, and the physical properties of van der Waals layered quasicrystals have never been studied to date. Here, we report on the discovery of superconductivity in a van der Waals layered quasicrystal of $\text{Ta}_{1.6}\text{Te}$. The electrical resistivity, magnetic susceptibility, and specific heat of the quasicrystal unambiguously validate the occurrence of bulk superconductivity at a transition temperature of ~ 1 K. This discovery can promote new research on assessing the physical properties of novel van der Waals layered quasicrystals as well as two-dimensional quasicrystals; moreover, it paves the way toward new frontiers of superconductivity in thermodynamically stable quasicrystals.

Introduction

In 1984, Shechtman et al. discovered an exotic solid phase in a rapidly quenched Al–Mn alloy whose electron diffraction pattern was sharp and exhibited ten-fold rotational symmetry¹. Soon after this discovery, a new classification scheme for solids was proposed, with the aforementioned phase denoted a quasicrystal (QC)^{2–4}, which was defined by quasiperiodic translational order and crystallographically forbidden rotational symmetries such as five-, ten-, and twelve-fold symmetries. In 1992, the International Union of Crystallography modified the definition of crystals in order to incorporate QCs into crystals⁵, and revised the classification scheme and definition of QCs accordingly. Now, all solids with long-range ordered atomic arrangements are called crystals. This includes ‘aperiodic crystals’, whose long-range order is not periodic but quasiperiodic. QCs are defined as aperiodic crystals that are neither incommensurate modulated structures nor aperiodic composite crystals. One significant difference between QCs and periodic crystals is that QCs are allowed to have a crystallographically forbidden rotational symmetry, which is, though, not a necessary but a sufficient condition for a QC in today’s context. Extensive studies over the last 40 years have shown that QCs are not exceptional but ubiquitous, as they have been formed in several binary and ternary systems. Of the 70 QCs reported to date, approximately 40 are thermodynamically stable⁶.

In 1998, Conrad et al. discovered a unique stable QC phase of Ta_{1.6}Te, which exhibited dodecagonal symmetry⁷. This new QC has several notable features that have not been observed to date. First, this phase is the only reported transition-metal chalcogenide QC. Second, this is the only QC of a van der Waals (vdW) layered material; its structure is realized through periodic stacking of two-dimensional dodecagonal QCs. The structures of this dodecagonal QC phase and the related crystal approximant (CA)

phases have been examined by X-ray diffractometry (XRD), electron diffractometry, and high-resolution transmission electron microscopy^{7–11}. Moreover, a few layers of the dodecagonal QC, which can be regarded virtually as a two-dimensional QC, have recently been isolated using a standard exfoliation technique⁸. However, the number of studies on this phase is still limited because conventional solidification techniques such as arc-melting and induction melting, which are usually used for the fabrication of QC alloys, cannot be applied since the melting temperature of Ta is much higher than the boiling temperature of Te. Moreover, experimental studies on elucidating the physical properties of the Ta_{1.6}Te dodecagonal QC phase have not been reported.

Recently, vdW layered crystals of transition-metal chalcogenides have attracted much attention and have been studied extensively owing to their fascinating physical properties^{12–16}. In the early stage of QC research, a few studies reported on the observations of superconductivity in icosahedral QCs (i-QCs) of Al–Cu–(Mg,Li)^{17,18}. However, convincing evidence for bulk superconductivity in QCs, indicated by zero resistivity, Meissner effect, and heat capacity jump, has only recently been presented; in 2018, Kamiya et al. observed the bulk superconductivity in a metastable i-QC of Al–Zn–Mg¹⁹. This i-QC phase, which was fabricated by a melt-spinning technique, exhibited a weak-coupling superconductivity at a transition temperature T_c of ~0.05 K. In the present study, we fabricate the Ta_{1.6}Te vdW layered QC phase by reaction sintering and measure the electrical resistivity, magnetic susceptibility, and specific heat down to low temperatures. The results categorically confirm the occurrence of bulk superconductivity in a thermodynamically stable QC for the first time at $T_c \approx 1$ K.

Results

Sample fabrication and characterization

Polycrystalline samples of the $\text{Ta}_{1.6}\text{Te}$ dodecagonal QC were fabricated by reaction sintering, which were powdered and then subjected to electron diffractometry and XRD analyses (see Methods section). The electron diffraction pattern of the synthesized sample (Fig. 1a) shows an arrangement of sharp spots with dodecagonal symmetry, verifying the formation of a dodecagonal QC phase. This pattern is essentially identical to that previously reported for the $\text{Ta}_{1.6}\text{Te}$ dodecagonal QC phase⁷. A powder XRD profile of the sample was acquired using $\text{Cu K}\alpha$ radiation (Fig. 1b); the $\text{Cu K}\alpha_2$ component was removed numerically, yielding a profile that corresponded to the $\text{Cu K}\alpha_1$ component (wavelength: 1.5405 Å). A comparison of this profile with those of the crystalline phases of TaTe_2 and Ta —the materials used for the reaction sintering—revealed no traces of these phases in the acquired profile, indicating the completion of the reaction and formation of another phase (Supplementary Fig. 1). However, because the powder XRD profile of the $\text{Ta}_{1.6}\text{Te}$ dodecagonal QC phase has not been previously reported, it was constructed based on that of the CA phase of $\text{Ta}_{97}\text{Te}_{60}$, whose structure was previously determined through single-crystal XRD measurements¹⁰. In general, by introducing phason strain to a QC structure, a CA structure can be obtained. Using this relationship, the powder XRD profile of a QC phase can be calculated from that of its CA phase. The principles and procedures underlying the calculation are detailed in Supplementary Section 1.2 (Supplementary Figs. 3–9 and Supplementary Tables 2 and 3). The acquired and constructed profiles were compared (Fig. 1b and the magnified profiles in Fig. 1c). Although the peaks in the acquired profile were not adequately resolved in the 2θ region of 33–47°, the overall agreement was satisfactory. Notably, comparing the acquired XRD profile with those of the $\text{Ta}_{21}\text{Te}_{13}$ and $\text{Ta}_{97}\text{Te}_{60}$ CA phases showed clear discrepancies

(Supplementary Fig. 9). The peak indices shown in Fig. 1c refer to the five basis vectors $\mathbf{a}_i^* = a^* (\cos(\frac{2\pi(i-1)}{12}), \sin(\frac{2\pi(i-1)}{12}), 0)$ ($i = 1-4$) and $\mathbf{c}^* = (0, 0, c^*)$, with a^* and c^* being 0.6942 \AA^{-1} and 0.6047 \AA^{-1} , respectively (Supplementary Fig. 2 and Supplementary Table 1). Complete data on the powder XRD profile calculated for the $\text{Ta}_{1.6}\text{Te}$ dodecagonal QC phase are provided in Supplementary Table 3.

Electrical resistivity

The temperature dependence of the electrical resistivity normalized by the value at $T = 300 \text{ K}$, $\rho/\rho_{300 \text{ K}}$ (where $\rho_{300 \text{ K}}$ is approximately $1.7 \text{ m}\Omega \text{ cm}$), is shown in Fig. 2a. The normalized electrical resistivity $\rho/\rho_{300 \text{ K}}$ gradually increased with decreasing temperature and abruptly descended to zero at a low temperature, indicating the emergence of superconductivity. Analysis of the temperature dependence of $\rho/\rho_{300 \text{ K}}$ below 2 K (Fig. 2b) indicated that $\rho(T)$ abruptly dropped to zero at a midpoint transition temperature $T_c^{\text{mid}} = 0.98 \text{ K}$, where $\rho(T_c^{\text{mid}}) = 0.5 \times \rho(1.1 \text{ K})$. Examination of the magnetic field dependence of $\rho/\rho_{300 \text{ K}}$ at different temperatures (Fig. 2c) suggests that the superconducting transitions were suppressed by increasing the magnetic field. Applying the criterion of 50% of the normal state resistivity to define the upper critical field H_{c2} , we evaluated H_{c2} and plotted the data against temperature (inset in Fig. 2d). This relation was analyzed (Fig. 2d) by plotting $-H_{c2}/[T_c(dH_{c2}/dT)_{T=T_c}]$ against T/T_c (that is, h^* vs. t). According to the Werthamer–Helfand–Hohenberg theory²⁰, which considers spin–orbit scattering and spin paramagnetism, H_{c2} in the dirty limit (mean free path $l \ll$ coherence length ξ_0) is expressed in terms of the digamma function as:

$$\ln \frac{1}{t} = \left(\frac{1}{2} + \frac{i\lambda_{\text{SO}}}{4\gamma} \right) \psi \left(\frac{1}{2} + \frac{\bar{h} + \frac{1}{2}\lambda_{\text{SO}} + i\gamma}{2t} \right) + \left(\frac{1}{2} - \frac{i\lambda_{\text{SO}}}{4\gamma} \right) \psi \left(\frac{1}{2} + \frac{\bar{h} + \frac{1}{2}\lambda_{\text{SO}} - i\gamma}{2t} \right) - \psi \left(\frac{1}{2} \right), \quad (1)$$

where ψ is the digamma function, $\gamma \equiv \left[(\alpha\bar{h})^2 - \left(\frac{1}{2}\lambda_{\text{SO}} \right)^2 \right]^{\frac{1}{2}}$, and $\bar{h} = \frac{4}{\pi^2} h^* = \frac{4H_{\text{c2}}}{\pi^2(-dH_{\text{c2}}/dT)_{t=1}}$, with parameters λ_{SO} and α representing the effects of the spin-orbit scattering and spin paramagnetism, respectively. The obtained data were adequately fit to equation (1) using $\alpha = \lambda_{\text{SO}} = 0$, indicating that the two effects were negligible. Through this fitting, the value of the zero-temperature upper critical field $H_{\text{c2}}(0)$ was deduced to be 32 kOe. The dirty-limit superconductivity was consistent with the large residual resistivity shown in Fig. 2a. A similar H_{c2} tendency has also been observed for the Al-Zn-Mg i-QC¹⁹.

Magnetic susceptibility and specific heat

The magnetic susceptibility and specific heat were investigated to corroborate the occurrence of bulk superconductivity. The temperature dependence of the magnetic susceptibility χ was monitored at an external magnetic field of 50 Oe under zero-field cooling (ZFC) and field cooling (FC) conditions (Fig. 3a). In the ZFC curve, a sharp drop occurred at an onset temperature of ≈ 0.95 K—a value consistent with the transition temperature corresponding to $\rho(T)$ in Fig. 2b; moreover, a large diamagnetism was observed, indicating the exclusion of the magnetic flux owing to the Meissner effect. The ZFC $4\pi\chi$ reached $-1.1 \text{ emu Oe}^{-1} \text{ cm}^{-3}$ at the lowest temperature (Fig. 3a), exceeding 100 % diamagnetization. This is because demagnetization-factor correction was not made due to the irregular shape of the sample, leading to an overestimation of $4\pi\chi$. A rough

estimation of the demagnetization factor N gave $N \approx 0.25$ (see Methods section). Then, the corrected $4\pi\chi$ at the lowest temperature was calculated to be $-0.86 \text{ emu Oe}^{-1} \text{ cm}^{-3}$. The ZFC data in a low temperature region is replotted in Fig. 3b, where a slope change is clearly evident at $T = 0.83 \text{ K}$ (see the temperature derivative curve in Supplementary Fig. 11b). This can be attributed to a superconducting transition of an impurity phase, which is most likely a high-order CA phase such as the previously reported CA phase of $\text{Ta}_{181}\text{Te}_{112}$ (ref. 8 and 9), which has a structure very similar to the dodecagonal QC phase. In any case, our ZFC $4\pi\chi$ data clearly indicates that the superconducting transition at $T = 0.95 \text{ K}$ emanates from the majority phase of the sample, which is the dodecagonal QC phase.

The temperature dependence of the specific heat C was probed by plotting C/T against T^2 (Fig. 3c). A jump in the C value was clearly observed at $T \approx 1 \text{ K}$ —a value consistent with the transition temperatures corresponding to $\rho(T)$ (Fig. 2b) and $\chi(T)$ (Fig. 3a). A linear behavior of C/T with T^2 of the form $C/T = \gamma + \beta T^2$ was observed at T^2 values above $\sim 1.4 \text{ K}^2$. Accordingly, the corresponding γ and β values were estimated as $0.0224 \text{ mJ g}^{-1} \text{ K}^{-2}$ and $0.00532 \text{ mJ g}^{-1} \text{ K}^{-4}$, respectively. The electronic part of the specific heat C_{el} was determined by subtracting the lattice contribution of the specific heat βT^3 from the measured specific heat, and $C_{\text{el}}/\gamma T$ was subsequently plotted against T (Fig. 3d). The jump height ΔC_{el} at T_c was $\Delta C_{\text{el}} \approx 1.0 \gamma T_c$, which was large enough to confirm the occurrence of bulk superconductivity.

With respect to the jump height ΔC_{el} , the Bardeen–Cooper–Schrieffer (BCS) theory predicts $\Delta C_{\text{el}} = 1.43 \gamma T_c$. Recently, Takemori et al.²¹ numerically studied the physical properties of quasiperiodic superconductors using an attractive Hubbard model on the Penrose tiling and showed $\Delta C_{\text{el}} = 1.21 \gamma T_c$. They demonstrated that this reduction in

ΔC_{el} is in agreement with the experimental results of the Al–Zn–Mg i-QC¹⁹. The specific heat data of the dodecagonal QC were compared with those of the Al–Zn–Mg i-QC and the theoretical curves (Fig. 4). The data of the dodecagonal QC also shows an obvious reduction in ΔC_{el} from the BCS value and are in rough agreement with those of the Al–Zn–Mg i-QC. Data at lower temperatures are necessary for more detailed comparisons.

Using the aforementioned value of β , the Debye temperature θ_D was deduced to be 132 K. Furthermore, the electron–phonon coupling constant λ_{ep} was obtained using the McMillan equation²²:

$$T_c = \frac{\theta_D}{1.45} \exp \left\{ - \frac{1.04(1 + \lambda_{ep})}{\lambda_{ep} - \mu^*(1 + 0.62\lambda_{ep})} \right\}, \quad (2)$$

where μ^* denotes the Coulomb pseudopotential parameter, which is typically considered 0.13 for superconductors including transition metals. Using $\mu^* = 0.13$, $T_c = 0.98$ K, and $\theta_D = 132$ K, λ_{ep} was calculated to be 0.52, indicating a weak-coupling superconductivity. In general, the density of a QC phase is approximately the same as that of its CA phase. Therefore, the density of the Ta_{1.6}Te dodecagonal QC phase was estimated to be 10.6 g cm⁻³, which was calculated for the Ta₉₇Te₆₀ CA phase from its structure data¹⁰. Using this value and the estimated values of γ and λ_{ep} , the density of states (for one of the two spin states) at the Fermi level $D(E_F)$ was estimated as 1.24×10^{47} states J⁻¹ m⁻³) using the expression²²:

$$D(E_F) = \frac{3\gamma}{2\pi^2 k_B^2 (1 + \lambda_{ep})}, \quad (3)$$

where k_B is the Boltzmann constant.

Discussion

Superconductivity in vdW layered crystals of tantalum dichalcogenides (TaCh_2 ; $\text{Ch}=\text{S}, \text{Se}, \text{and Te}$) has been extensively studied in recent years^{23–32}. Superconductivity in this group of materials often occurs in competition with a charge density wave (CDW) state. In TaTe_2 , a CDW state is stable down to the lowest temperature at ambient pressure without transitioning to a superconducting state; superconductivity appears only when the CDW state is suppressed, e.g., by applied pressure²³. In $2H\text{-TaS}_2$, $4H_b\text{-TaS}_2$ and $2H\text{-TaSe}_2$, superconductivity develops in the temperature range below that of the CDW state at ambient pressure. For $4H_b\text{-TaS}_2$, on the other hand, the realization of chiral superconductivity has recently been suggested²⁴.

The two-dimensionality of superconductivity is one of the major issues in these materials. The anisotropy parameter γ defined as $\gamma = H_{c2}^{\parallel}(0)/H_{c2}^{\perp}(0)$ is approximately 4.5, 17, and 2.8 for $2H\text{-TaS}_2$, $4H_b\text{-TaS}_2$, and $2H\text{-TaSe}_2$, respectively^{25–27}. In $4H_b\text{-TaS}_2$, the degree of two-dimensionality is considerably high. This is due to the alternating stacking structure of superconducting $1H$ and Mott insulating $1T$ layers. In $2H\text{-TaS}_2$ and $2H\text{-TaSe}_2$, the anisotropy is not so strong and the superconductivity is essentially three-dimensional with weak anisotropy. However, the degree of two-dimensionality can be further enhanced by weakening the interlayer coupling, for instance, through intercalation. Quasi-two-dimensional superconductivity has been realized in $2H\text{-TaS}_2$ samples intercalated with Pb ²⁸ and pyridine²⁹. Another direct method to realize two-dimensional superconductivity is to synthesize a very thin sample, for example, by exfoliation. Atomically thin $2H\text{-TaS}_2$ and $2H\text{-TaSe}_2$ have been shown to exhibit significantly higher T_c than the corresponding bulk values^{30,31}. In the absence of in-plane inversion symmetry, Zeeman-type spin–orbit interaction forces the electron spins to align perpendicular to the plane in the two-dimensional limit. The realization of such an ‘Ising superconductivity’

has been reported for monolayer $1H\text{-TaS}_2$ ³². In this context, the elucidation of the two-dimensional nature of superconductivity in the vdW layered quasicrystal of $\text{Ta}_{1.6}\text{Te}$ is an important and challenging task, which will be achieved by the fabrication of sizable single-crystal samples, direction-dependent magnetic and transport measurements, the fabrication of intercalated and exfoliated samples, and their direction-dependent measurements.

The superconducting parameters of the Ta–chalcogenides^{24,33,34} including Ta-rich Ta_2Se and $\text{Ta}_{1.6}\text{Te}$ dodecagonal QC (from the present study), together with those of Ta³⁵, and Al–Zn–Mg i-QC¹⁹ are summarized in Table 1. For a simple metal system of the Al–Zn–Mg i-QC, $\mu^* = 0.1$ was used²² in the calculation of λ_{ep} , while for the others $\mu^* = 0.13$ was used. The λ_{ep} values of Ta and Ta_2Se were fairly high, which may be in an intermediate-coupling range. The other systems exhibited sufficiently small λ_{ep} values appropriate for weak-coupling superconductors, with the λ_{ep} value of the Al–Zn–Mg i-QC being considerably smaller than those of the others. Also for $D(E_F)$, the Al–Zn–Mg i-QC exhibited the smallest value among the specimens. For layered Ta–chalcogenides, θ_D scaled with $M_{\text{ave}}^{-1/2}$ (M_{ave} : average atomic weight) except for Ta_2Se , as shown in Supplementary Discussion 2 (Supplementary Fig. 10).

Recent theoretical studies^{36–38} have revealed that quasicrystalline superconductors exhibit several unconventional behaviors that are typically not observed in other known superconductors in periodic and disordered systems, thus opening a new field in the research of superconductivity. Nagai³⁶ has studied superconducting tight-binding models of Penrose and Ammann–Beenker lattices (typical two-dimensional quasicrystalline lattices) and demonstrated an intrinsic vortex pinning due to spatially inhomogeneous

superconducting order parameter. Such an inhomogeneous order parameter arises from the quasicrystalline structural order, and therefore, the vortex pinning occurs without an impurity or defect. Sakai et al.³⁷ have investigated quasicrystalline superconductivity using an attractive Hubbard model on a Penrose lattice using the real-space dynamical mean-field theory. Unconventional spatially-extended Cooper pairs were formed; the sum of the momenta of the Cooper pair electrons was nonzero, in contrast to the zero total momentum of the Cooper pair in the conventional BCS superconductivity. Such a nonzero total momentum of the Cooper pair is also observed for the Fulde–Ferrell–Larkin–Ovchinnikov (FFLO) state previously proposed for periodic systems^{39–42}. However, the unconventional Cooper pairing in the model QC is completely different from the FFLO state because the Cooper pairing occurs under no magnetic field. In addition, under a high magnetic field, a state similar to the FFLO state is formed in the model QC³⁸. However, this state is also different from the conventional FFLO state in periodic systems and forms a fractal-like spatial pattern of the oscillating superconducting order parameter, which is compatible with the self-similar structural order that is possessed by the QCs. As mentioned above, many interesting features are theoretically expected for superconducting QCs, which are yet to be demonstrated experimentally, and the Ta_{1.6}Te dodecagonal QC phase in the present study offers a precious platform for it.

In conclusion, polygrain Ta_{1.6}Te dodecagonal QC samples were fabricated by reaction sintering. Careful phase identification of the sample was performed by electron and powder X-ray diffraction experiments and diffraction-profile simulations. The samples were subjected to electrical resistivity, magnetic susceptibility, and specific heat measurements. The results unconditionally validate the occurrence of bulk superconductivity at a T_c of ~ 1 K. This is the first example of superconductivity in

thermodynamically stable QCs. These findings are expected to motivate further investigations into the physical properties of vdW layered quasicrystals as well as two-dimensional quasicrystals. In particular, the dodecagonal QC provides a valuable platform for the experimental demonstration of the unique superconductivity theoretically predicted for QCs.

Methods

Sample synthesis

Ta_{1.6}Te samples were synthesized by reaction sintering. TaTe₂ and Ta were used as the starting materials at a ratio of 1:3. TaTe₂ was prepared in advance from a 1:2 elemental powder mixture of Ta (~325 mesh, 99.9%; Rare Metallic) and Te (~100 mesh, 99.99%; Rare Metallic). The mixture was pressed into a pellet (diameter of 10 mm) and inserted into a quartz tube, which was evacuated to $5-6 \times 10^{-4}$ Pa, sealed, and then subjected to two-step annealing (773 K for 24 h, followed by 1,273 K for 24 h). Subsequently, the mixture of the TaTe₂ powder, Ta powder, and a shot (30–40 mg) of iodine (99.9%; FUJIFILM Wako Pure Chemical; used to promote the reaction) was pressed into a 5-mm-thick pellet (diameter of 10 mm). The pellet was sealed in an evacuated ($5-6 \times 10^{-4}$ Pa) quartz tube and sintered at 1,273 K for six days.

Sample characterization

The structures of the sintered samples were examined by powder X-ray diffractometry (XRD) and electron diffractometry. Powder XRD profiles were obtained using a diffractometer (RINT-2500V; Rigaku) with Cu K α radiation at 40 kV and 200 mA, a divergence slit open-angle of 0.5°, scattering slit open-angle of 0.5°, receiving slit width

of 0.15 mm, rate of $0.5^\circ \text{ min}^{-1}$, and step of 0.01° . Electron diffraction patterns were obtained using a transmission electron microscope (JEM-2010F; JEOL) operated at an acceleration voltage of 200 kV.

Physical property measurements

The electrical resistivity was measured using a physical property measurement system (PPMS; Quantum Design) with a four-probe method and a direct current of 25–100 μA . Samples with cross-sectional areas of 2–3 mm \times 2–3 mm and lengths of 5–6 mm were used for the electrical resistivity measurements. The magnetic field was applied in the direction roughly perpendicular to the electrical current. The sample for magnetic susceptibility measurements was prepared by crushing the sintered pellet. A piece of 95.64 mg was subjected to the magnetic susceptibility measurements using a magnetic property measurement system (MPMS3; Quantum Design) equipped with a ^3He refrigerator. Using the density (10.6 g cm^{-3}) calculated for the CA phase¹⁰, which should be approximately the same as the density of the dodecagonal QC phase, the volume was estimated to be 9.02 mm^3 . The ratio of the typical length of the sample along z -direction to that in xy -plane ($z//\mathbf{H}$) was roughly 1.5, giving the demagnetization factor of $N \approx 0.25$ (ref. 43). The sample for specific heat measurements was prepared also by crushing the sintered pellet. A piece was picked up and roughly shaped and polished with sandpaper so that the side to be placed on the sample platform is flat. A sample with a weight of 26.09 mg and the size of $\sim 2.2 \times 2.2 \times 0.5 \text{ mm}^3$ was subjected to the specific heat measurements using the PPMS with the thermal relaxation method.

Data availability

The data that support the findings of this study are available within the paper and its Supplementary Information. Source data are provided with this paper.

References

1. Shechtman, D., Blech, I., Gratias, K. & Cahn, J. W. Metallic phase with long-ranged orientational order and no translational symmetry. *Phys. Rev. Lett.* **53**, 1951–1953 (1984).
2. Levine, D. & Steinhardt, P. J. Quasicrystals: a new class of ordered structures. *Phys. Rev. Lett.* **53**, 2477–2480 (1984).
3. Levine, D. & Steinhardt, P. J. Quasicrystals I: definition and structure. *Phys. Rev. B* **34**, 596–616 (1986).
4. Socolar, J. E. S. & Steinhardt, P. J. Quasicrystals II: unit-cell configurations. *Phys. Rev. B* **34**, 617–647 (1986).
5. International Union of Crystallography, Report of the Executive Committee for 1991. *Acta Cryst.* **A48**, 922–946 (1992).
6. Tsai, A. P. Discovery of stable icosahedral quasicrystals: progress in understanding structure and properties. *Chem. Soc. Rev.* **42**, 5352–5365 (2013).
7. Conrad, M., Krumeich, F. & Harbrecht, B. A dodecagonal quasicrystalline chalcogenide. *Angew. Chem. Int. Ed.* **37**, 1383–1386 (1998).
8. Cain, J. D., Azizi, A., Conrad, M. & Zettl, A. Layer-dependent topological phase in a two-dimensional quasicrystal and approximant. *Proc. Natl. Acad. Sci. U. S. A.* **117**, 26135–26140 (2020).
9. Conrad, M., Krumeich, F., Reich, C. & Harbrecht, B. Hexagonal approximants of a dodecagonal tantalum telluride – the crystal structure of Ta₂₁Te₁₃. *Mater. Sci. & Eng.* **294-296**, 37–40 (2000).
10. Conrad, M. & Harbrecht, B. Ta₉₇Te₆₀: a crystalline approximant of a tantalum telluride quasicrystal with twelvefold rotational symmetry. *Chem. Eur. J.* **8**, 3093–3102 (2002).

11. Krumeich, F., Conrad, M., Nissen, H.-U. & Barbrecht, B. The mesoscopic structure of disordered dodecagonal tantalum telluride: a high-resolution transmission electron microscopy study. *Phil. Mag. Lett.* **78**, 357–367 (1998).
12. Deguchi, K., Takano, Y. & Mizuguchi, Y. Physics and chemistry of layered chalcogenide superconductors. *Sci. Technol. Adv. Mater.* **13**, 054303 (2012).
13. Wang, P., Yang, Y., Pan, E., Liu, F., Ajayan, P. M., Zhou, J. & Liu, Z. Emerging phases of layered metal chalcogenides. *Small* **18**, 2105215 (2122).
14. Duong, D. L., Yun, S. J. & Lee, Y. H. Van der Waals Layered materials: opportunities and challenges. *ACS Nano* **11**, 11803–11830 (2017).
15. Tan, C., Cao, X., Wu, X.-J., He, Q., Yang, J., Zhang, X., Chen, J., Zhao, W., Han, S., Nam, G.-H., Sindoro, M. & Zhang, H. Recent advances in ultrathin two-dimensional nanomaterials. *Chem. Rev.* **117**, 6225–6331 (2017).
16. Yi, Y., Chen, Z., Yu, X.-F., Zhou, Z.-K. & Li, J. Recent advances in quantum effects of 2D materials. *Adv. Quantum Technol.* **2**, 1800111 (2019).
17. Wong, K. M., Lopdrup, E., Wagner, J. L., Shen, Y. & Poon, S. J. Transport and superconducting properties of the $\text{Mg}_{32}(\text{Al,Zn})_{49}$ -type quasicrystalline and crystalline phases. *Phys. Rev. B* **35**, 2494–2497 (1987).
18. Wagner, J. L., Biggs, B. D., Wong, K. M. & Poon, S. J. Specific-heat and transport properties of alloys exhibiting quasicrystalline and crystalline order. *Phys. Rev. B* **38**, 7436–7441 (1988).
19. Kamiya, K., Takeuchi, T., Kabeya, N., Wada, N., Ishimasa, T., Ochiai, A., Deguchi, K., Imura, K. & Sato, N. K. Discovery of superconductivity in quasicrystal. *Nat. Commun.* **9**, 154 (2018).
20. Werthamer, N. R., Helfand, K. & Hohenberg, P. C. Temperature and purity

- dependence of the superconducting critical field, H_{c2} . III. Electron spin and spin-orbit effects. *Phys. Rev.* **147**, 295–302 (1966).
21. Takemori, N., Arita, R. & Sakai, S. Physical properties of weak-coupling quasiperiodic superconductors. *Phys. Rev. B* **102**, 115108 (2020).
 22. McMillan, W. L. Transition temperature of strong-coupled superconductors. *Phys. Rev.* **167**, 331–334 (1968).
 23. Guo, J., Huang, C., Luo, H., Yang, H., Wei, L., Cai, S., Zhou, Y., Zhao, H., Li, X., Li, Y., Yang, K., Li, A., Sun, P., Li, J., Wu, Q., Cava, R. J. & Sun, L. Observation of three superconducting transitions in the pressurized CDW-bearing compound TaTe₂. *Phys. Rev. Mater.* **6**, L051801 (2022).
 24. Ribak, A., Skiff, R. M., Mograbi, M., Rout, P. K., Fischer, M. H., Ruhman, J., Chashka, K., Dagan, Y. & Kanigel, A. Chiral superconductivity in the alternate stacking compound 4Hb-TaS₂. *Sci. Adv.* **6**, eaax9480 (2020).
 25. Abdel-Hafiez, M., Zhao, X. M., Kordyuk, A. A., Fang, Y. W., Pan, B., He, Z., Duan, C. G., Zhao, J. & Chen, X. J. Enhancement of superconductivity under pressure and the magnetic phase diagram of tantalum disulfide single crystals. *Sci. Rep.* **6**, 31824 (2016).
 26. Xie, Z., Yang, M., Cheng, Z. G., Ying, T., Guo, J. & Chen, X. A revisit of superconductivity in 4H_b-TaS_{2-2x}Se_{2x} single crystals. *J. Phys. Soc. Jpn.* **92**, 054702 (2023).
 27. Yokota, K., Kurata, G., Matsui, T. & Fukuyama, H. Superconductivity in the quasi-two-dimensional conductor 2H-TaSe₂. *Physica B* **284–288**, 551–552 (2000).
 28. Yang, X., Yu, T., Xu, C., Wang, J., Hu, W., Xu, Z., Wang, T., Zhang, C., Ren, Z., Xu, Z., Hirayama, M., Arita, R. & Lin, X. Anisotropic superconductivity in the

- topological crystalline metal $\text{Pb}_{1/3}\text{TaS}_2$ with multiple Dirac fermions. *Phys. Rev. B* **104**, 035157 (2021).
29. Coleman, R. V., Eiserman, G. K., Hillenius, S. J., Mitchell, A. T. & Vicent, J. L. Dimensional crossover in the superconducting intercalated layer compound $2H\text{-TaS}_2$. *Phys. Rev. B* **27**, 125 (1983).
30. Navarro-Moratalla, E., Island, J. O., Mañas-Valero, S., Pinilla-Cienfuegos, E., Castellanos-Gomez, A., Quereda, J., Rubio-Bollinger, G., Chirolli, L., Silva-Guillén, J. A., Agraït, N., Steele, G. A., Guinea, F., van der Zant H. S. J. & Coronado, E. Enhanced superconductivity in atomically thin TaS_2 . *Nat. Commun.* **7**, 11043 (2016).
31. Wu, Y., He, J., Liu, J., Xing, H., Mao, Z. & Liu, Y. Dimensional reduction and ionic gating induced enhancement of superconductivity in atomically thin crystals of $2H\text{-TaSe}_2$. *Nanotechnology* **30**, 035702 (2019).
32. de la Barrera, S. C., Sinko, M. R., Gopalan, D. P., Sivadas, N., Seyler, K. L., Watanabe, K., Taniguchi, T., Tsen, A. W., Xu, X., Xiao, D. & Hunt, B. M. Tuning Ising superconductivity with layer and spin-orbit coupling in two-dimensional transition-metal dichalcogenides. *Nat. Commun.* **9**, 1427 (2018).
33. Gui, X., Górnicka, K., Chen, Q., Zhou, H., Klimczuk, T. & Xie, W. Superconductivity in metal-rich chalcogenide Ta_2Se . *Inorg. Chem.* **59**, 5798–5802 (2020).
34. Harper, J. M. E., Geballe, T. H. & DiSalvo, F. J. Thermal properties of layered transition-metal dichalcogenides at charge-density-wave transitions. *Phys. Rev. B* **15**, 2943–2951 (1977).
35. Worley, R. D., Zemansky, M. W. & Boorse, H. A. Heat Capacities of vanadium and tantalum in the normal and superconducting phases. *Phys. Rev.* **99**, 447–458 (1955).
36. Nagai, Y. Intrinsic vortex pinning in superconducting quasicrystals. *Phys. Rev. B* **106**,

064506 (2022).

37. Sakai, S., Takemori, N., Koga, A. & Arita, R. Superconductivity on a quasiperiodic lattice: extended-to-localized crossover of Cooper pairs. *Phys. Rev. B* **95**, 024509 (2017).
38. Sakai, S. & Arita, R. Exotic pairing state in quasicrystalline superconductors under a magnetic field. *Phys. Rev. Res.* **1**, 022002(R) (2019).
39. Fulde, P. & Ferrell, R. A. Superconductivity in a strong spin-exchange field. *Phys. Rev.* **135**, A550–563 (1964).
40. Larkin, A. & Ovchinnikov, Y. Nonuniform state of superconductors. *Zh. Eksp. Teor. Fiz.* **47**, 1136–1146 (1964) [*Sov. Phys. JETP* **20**, 762–770 (1965)].
41. Casalbuoni, R. & Nardulli, G. Inhomogeneous superconductivity in condensed matter and QCD. *Rev. Mod. Phys.* **76**, 263–320 (2004).
42. Matsuda, Y. & Shimahara, H. Fulde–Ferrell–Larkin–Ovchinnikov state in heavy fermion superconductors. *J. Phys. Soc. Jpn.* **76**, 051005 (2007).
43. Sato, M. & Ishii, Y. Simple and approximate expressions of demagnetizing factors of uniformly magnetized rectangular rod and cylinder. *J. Appl. Phys.* **66**, 983–985 (1989).

Acknowledgements

This study was supported by the JST-CREST program (grant no. JPMJCR22O3; Japan) (R.T. and K.E.) and JSPS KAKENHI Grant Numbers JP19H05821 (R.T. and K.E.) and JP23K04355 (Y.T.). A part of this work was performed using facilities of the Cryogenic Research Center, the University of Tokyo, and Advanced Research Infrastructure for Materials and Nanotechnology in Japan (ARIM) of the Ministry of Education, Culture, Sports, Science and Technology (MEXT), grant JPMXP1222UT0053.

Author contributions

K.H., S.N., and Y.T. fabricated the samples and performed powder XRD measurements. S.N., Y.K. and K.E. analyzed the powder XRD data and constructed the profile. Y.K., K.H., and Y.T. performed the electron diffraction experiments. Y.T., K.H., and K.E. conducted the electrical resistivity and specific heat measurements. S.S., R.T., Y.T., and K.H. conducted the magnetic susceptibility experiments. K.E. designed and supervised the whole project. Y.T. and K.E. cowrote the manuscript. All the authors have discussed the results and contributed to the manuscript.

Competing interests

The authors declare no competing interests.

Additional information

Supplementary information The online version contains supplementary material available at <link>.

Correspondence and requests for materials should be addressed to Yuki Tokumoto or Keiichi Edagawa.

Reprints and permissions information is available at www.nature.com/reprints.

Table 1 | Superconducting characteristics of the Ta_{1.6}Te dodecagonal quasicrystal (dd-QC), other Ta–chalcogenide systems, Ta, and the Al–Zn–Mg i-QC. T_c and θ_D were sourced from the literature, except for 4H_b-TaS₂. T_c and θ_D for 4H_b-TaS₂ were obtained by fitting the specific heat data presented in the literature. λ_{ep} was calculated from T_c and θ_D using equation (2), where μ^* values of 0.1 and 0.13 were used for the Al–Zn–Mg i-QC and the other systems, respectively. $D(E_F)$ was calculated from λ_{ep} and reported γ using equation (3).

	T_c (K)	θ_D (K)	λ_{ep}	$D(E_F)$ (10^{47} states J ⁻¹ m ⁻³)	References
Ta _{1.6} Te dd-QC	0.98	132	0.52	1.24	Present study
Ta ₂ Se	3.8	271	0.61	1.82	[33]
2H-TaSe ₂	0.15	202	0.37	0.685	[34]
2H-TaS ₂	0.8	236	0.45	1.19	[34]
4H _b -TaS ₂	2.7	241	0.57	0.805	[24]
Ta	4.4	231	0.67	2.39	[35]
Al–Zn–Mg i-QC	0.05	317	0.27	0.453	[19]

Fig. 1 | Electron and X-ray diffraction data. **a**, Electron diffraction pattern of the synthesized $\text{Ta}_{1.6}\text{Te}$. **b**, Experimentally obtained powder XRD profile of the synthesized $\text{Ta}_{1.6}\text{Te}$ and the calculated profile for the dodecagonal QC phase. **c**, Magnified version of the data shown in **b**. The principles and procedures underlying the calculations are detailed in Supplementary Section 1.2.

Fig. 2 | Electrical resistivity measurements. **a**, Temperature dependence of the electrical resistivity normalized by the value at $T = 300$ K, i.e., $\rho/\rho_{300\text{K}}$. **b**, Temperature dependence of $\rho/\rho_{300\text{K}}$ magnified at temperatures below 2 K. **c**, Magnetic field dependence of $\rho/\rho_{300\text{K}}$ at different temperatures. **d**, Temperature dependence of the upper critical field H_{c2} analyzed by plotting $h^* = -H_{c2}/[T_c(dH_{c2}/dT)_{T=T_c}]$ against $t = T/T_c$. Open squares represent the experimentally obtained data, and the dashed curve represents the Werthamer–Helfand–Hohenberg fit with $\alpha = 0$ and $\lambda_{SO} = 0$. The inset shows the temperature dependence of H_{c2} .

Fig. 3 | Magnetic susceptibility and specific heat measurements. **a**, Temperature dependence of the magnetic susceptibility χ . **b**, The same as **a**, but in a low temperature region. **c**, Temperature dependence of specific heat C , analyzed by plotting C/T against T^2 ; and **d**, the ratio between the electronic component of the specific heat C_{el} and γT .

Fig. 4 | Temperature dependences of specific heats. The experimental data of the dodecagonal QC (black dots) and the Al-Zn-Mg i-QC (red dots), and the data numerically obtained for the Penrose tiling (blue dots) and the curve predicted by the BCS theory (black line). The data for the Al-Zn-Mg i-QC and Penrose tiling were extracted from the figures in the original papers^{19,21}.